# JAX Animal Behavior System (JABS), a genetics-informed, end-to-end advanced behavioral phenotyping platform for the laboratory mouse

Anshul Choudhary[†], Brian Q Geuther[†], Thomas J Sproule[†], Glen Beane[†], Vivek Kohar[‡], Jarek Trapszo, Vivek Kumar*

The Jackson Laboratory, Bar Harbor, United States

## eLife Assessment

This **important** study presents JABS, an open-source platform that integrates hardware and user-friendly software for standardized mouse behavioral phenotyping. The work has practical implications for improving reproducibility and accessibility in behavioral neuroscience, especially for linking behavior to genetics across diverse mouse strains. The strength of evidence is **convincing**, with comprehensive validation of the platform's components and enthusiastic reviewer support.

**\*For correspondence:**
Vivek.Kumar@jax.org

[†]These authors contributed equally to this work

**Present address:** [‡]UniBio Intelligence, Bar Harbor, United States

## Abstract

Automated detection of complex animal behavior remains a challenge in neuroscience. Developments in computer vision have greatly advanced automated behavior detection and allow high-throughput preclinical and mechanistic studies. An integrated hardware and software solution is necessary to facilitate the adoption of these advances in the field of behavioral neurogenetics, particularly for non-computational laboratories. We have published a series of papers using an open field arena to annotate complex behaviors such as grooming, posture, and gait as well as higher-level constructs such as biological age and pain. Here, we present our integrated rodent phenotyping platform, JAX Animal Behavior System (JABS), to the community for data acquisition, machine learning-based behavior annotation and classification, classifier sharing, and genetic analysis. The JABS Data Acquisition Module (JABS-DA) enables uniform data collection with its combination of 3D hardware designs and software for real-time monitoring and video data collection. JABS-Active Learning Module (JABS-AL) allows behavior annotation, classifier training, and validation. We introduce a novel graph-based framework (*ethograph*) that enables efficient boutwise comparison of JABS-AL classifiers. JABS-Analysis and Integration Module (JABS-AI), a web application, facilitates users to deploy and share any classifier that has been trained on JABS, reducing the effort required for behavior annotation. It supports the inference and sharing of the trained JABS classifiers and downstream genetic analyses (heritability and genetic correlation) on three curated datasets spanning 168 mouse strains that we are publicly releasing alongside this study. This enables the use of genetics as a guide to proper behavior classifier selection. This open-source tool is an ecosystem that allows the neuroscience and genetics community to share advanced behavior analysis and reduces the barrier to entry into this new field.

## Introduction

Behavioral analysis in animal models seeks to link complex and dynamic behaviors with underlying genetic and neural circuit functions (*Gomez-Marin et al., 2014*). In the context of disease, altered

genetic circuits shape altered neural circuits, which in turn produce altered behaviors. The primary purpose of behavior analysis in the animal is to understand the mechanisms of disease and to seek novel therapeutics to improve human health. The laboratory mouse has been at the forefront of these discoveries. However, linking altered genetic circuits to functional changes in neural circuits and ultimately behavior is challenging. These challenges are broad; however, one major hurdle has always been a behavior quantification task itself. Animal behavior quantification has rapidly advanced in the past few years with the application of machine learning to the problem of behavior annotation and with the adoption of computational ethology approaches to behavioral neurogenetics (*Datta et al., 2019*; *Pereira et al., 2020*; *Luxem et al., 2023*; *Anderson and Perona, 2014*; *Mathis et al., 2020*).

These advances are mainly due to breakthroughs in the statistical learning and computer science fields which have been adopted and extended for biological applications and have made the task of behavior annotation at high resolution scalable and objective, with increased accuracy (*Choi and Kumar, 2024*). Although significant advances have been made in the annotation of animal behavior using machine vision, a major challenge remains in the democratization of these technologies. As a simple example, many labs adopt their existing apparatus to generate an intermediate representation of the animal for tasks such as tracking. These are often segmentation masks or keypoints. Each lab generally trains a custom model for these, which, depending on the complexity of the task, can require large amounts of human-annotated training data. Many do not validate or even report the performance of their models, which are taken at face value to work. This is a large data labeling burden that is repeated by individual labs. The next step of extracting behaviors from these intermediate representations is even more challenging. The process entails creating features from intermediate representations followed by heuristics or classifiers to determine when a behavior of interest occurs. Behaviorists often disagree on behavior definition, even within labs, and therefore these behavior classifiers are incredibly valuable. They encode a behaviorist's expertise in the form of mathematical weights. Since labs start with niche behavior apparatus and intermediate representations, the process of feature extraction, classification, and the logic of assigning behaviors stays within a lab. That is, it is challenging for labs to share classifiers, because they only work in their hardware setup. This paradigm is not sustainable and prohibits the application of engineering principles to biology. The paradigm described above, combined with the fact that a high level of expertise is needed for proper use and interpretation of machine learning methods, can be a challenge to the reproducibility and replicability of scientific discoveries and ultimately therapeutic discoveries.

With this in mind, we present two complementary systems that are designed for behavior characterization in rodent models. The first platform, called JAX Animal Behavior System (JABS), consists of video collection hardware and software, a behavior labeling and active learning app, and an online database for sharing classifiers. This is an open field system which we have used in over 6 papers (*Geuther et al., 2019*; *Geuther et al., 2021*; *Sheppard et al., 2022*; *Hession et al., 2021*; *Guzman et al., 2023*; *Sabnis et al., 2022*). Adoption of JABS will allow laboratories to bypass the need for creating segmentation or pose estimation models for routine open-field tasks. In addition, existing models for frailty (*Hession et al., 2022*), nociception (*Sabnis et al., 2022*), seizures (*Sabnis et al., 2024*), and others can be adopted. The second, called Digital InVivo System (DIV Sys), is hardened and scalable home cage monitoring system (see *Robertson et al., 2024*). Both end-to-end systems are designed to enable community members to leverage others' work and to extend the capabilities of the system. We hope that these platforms will be adopted and extended by the community.

JABS adds to a list of commercial and open source animal tracking platforms. JABS covers hardware, behavior prediction, a shared resource for classifiers, and genetic association studies. We are not aware of another system that encompasses all these components. Commercial packages such as EthoVision XT and HomeCage Scan give users a ready-made camera-plus-software solution that automatically tracks each mouse and reports simple measures such as distance traveled or time spent in preset zones, but they do not provide open hardware designs, editable behavior classifiers, or any genetics workflow. At the open-source end, there are greater than 100 projects cataloged on Open-Behavior and summarized in recent reviews (*Luxem et al., 2023*; *Isik and Unal, 2023*) that usually cover only one link in the chain DIY rigs, pose-tracking libraries (e.g. DeepLabCut *Mathis et al., 2018*, SLEAP *Pereira et al., 2022*) or supervised and unsupervised behavior-classifier pipelines (e.g. SimBA *Goodwin et al., 2024*, MARS *Segalin et al., 2021*, JAABA *Kabra et al., 2013*, B-SOiD *Hsu and Yttri, 2021*, DeepEthogram *Bohnslav et al., 2021*). JABS provides an open source ecosystem

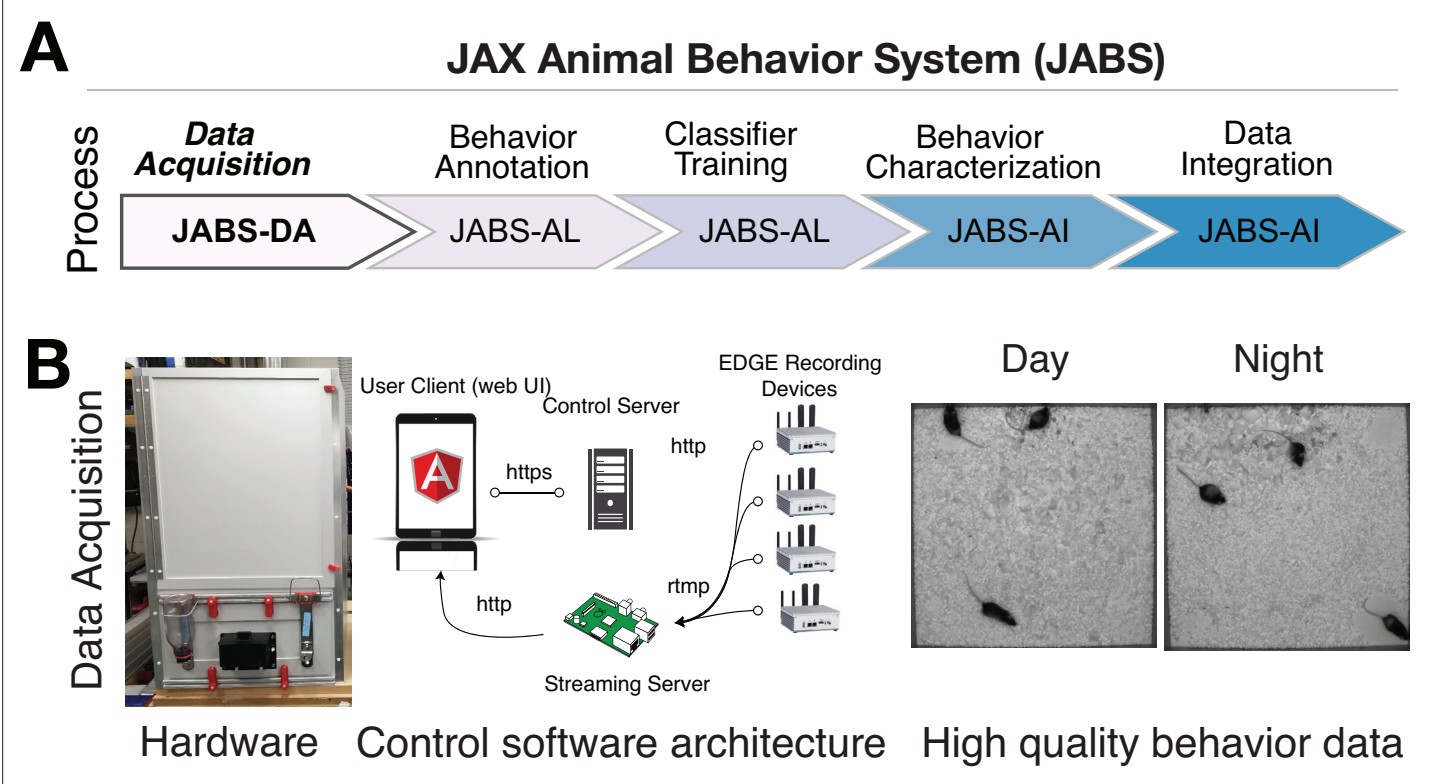

**Figure 1.** JABS data acquisition module (JABS-DA). The JABS-DA consists of hardware and software for video data acquisition and processing. (**A**) JABS pipeline highlighting individual steps towards automated behavioral quantification. (**B**) Detailed example of JABS data acquisition, including a picture of the monitoring hardware, architecture of the real-time monitoring app, and screenshots from videos taken during daytime and nighttime. The open field arena is shown from the outside (left), and a screenshot of video data is shown on the right. JABS-DA blocks visible light to the camera and only collects data using IR illumination, which produces uniform data during day and night. The JABS-DA computer hardware and software (middle) allow streaming of video data from edge devices, which enables remote welfare checks and web-based experiment setup and monitoring. Data compression is handled on these edge devices.

that integrates all four: (i) top-down arena hardware with parts list and assembly guide; (ii) an active-learning GUI that produces shareable classifiers; (iii) a public web service that enables sharing of the trained classifier and applies any uploaded classifier to a large and diverse strain survey; and (iv) built-in heritability, genetic-correlation, and GWAS reporting.

## Results

JABS is an integrated platform developed in our lab over the past 5 years with pose and segmentation models as intermediate representations. Our lab has previously used computer-vision methods to track visually diverse mice under different environmental conditions (*Geuther et al., 2019*), infer pose for gait analysis and posture analysis (*Sheppard et al., 2022*), and detect complex behaviors like grooming (*Geuther et al., 2021*). We have also used computer vision-derived features to predict complex constructs, such as health and frailty and pain (*Hession et al., 2021*; *Sabnis et al., 2022*). These models have been trained and validated on genetically diverse mouse strains for high-quality foundational metrics (*Geuther et al., 2019*; *Sheppard et al., 2022*). JABS hardware and software has been used to characterize complex behaviors such as grooming, gait, posture, as well as complex states such as frailty, pain, and intensity of seizures (*Geuther et al., 2019*; *Geuther et al., 2021*; *Sheppard et al., 2022*; *Hession et al., 2021*). JAX has made components of JABS, including ML models, free to use for non-commercial purposes.

The process and various components of JABS are illustrated in *Figure 1A*. Briefly, our system comprises three components encompassing five different processes, namely, (i) data acquisition, (ii) behavior annotation, (iii) classifier training, (iv) behavior characterization, and (v) data integration. The

first component (JABS-DA module) is the custom-designed standardized data acquisition hardware and software that provides a controlled environment, optimized video storage, and live monitoring capabilities. The second component (JABS-AL module) is a Python-based GUI active learning app for behavior annotation and training classifiers using the annotated data. One can then use the trained classifiers for predicting whether behavior happens or not in the unlabeled frames. The last component of JABS is the analysis and integration module (JABS-AI), a web application that provides an interactive user interface to browse through the strain survey results from different classifiers, download existing classifiers and related training data. The app can also be used to classify various behaviors in user-submitted videos (pose files) using the classifiers available in the database. Furthermore, researchers have the option to contribute their custom classifiers, trained through the JABS-AL app. These user-generated classifiers can be submitted to perform predictions within our extensive strain survey dataset, coupled with comprehensive genetic analysis, including assessments of heritability and genetic correlations. Next, we discuss the individual components of JABS in detail.

## JABS data acquisition - hardware and software

We use a standardized hardware setup for high-quality data collection and optimized storage (*Figure 1B*). The end result is uniform video data across day and night. Complete details of the software and hardware, including 3D designs used for data collection, are available on our Github (https://github.com/KumarLabJax/JABS-data-pipeline copy archived at *Geuther et al., 2026a*). We also provide a step-by-step assembly guide (https://github.com/KumarLabJax/JABS-data-pipeline/blob/main/Multi-day%20setup%20PowerPoint%20V3.pptx).

We have organized the animal habitat design into three groups of specifications. The first group of specifications is requirements necessary for enabling compatibility with our machine learning algorithms. The second group describes components that can be modified as long as they produce data that adheres to the first group. The third group describes components that do not affect compatibility with our machine learning algorithms. While we distinguish between abstract requirements in group 1 and specific hardware in group 2 that meet those requirements, we recommend that users of our algorithms use our specific hardware in group 2 to ensure compatibility.

The design elements that are critical to match specifications in order to re-use machine learning algorithms include (1a) the camera viewpoint, (1b) minimum camera resolution and frame rate, (1c) field of view and imaging distance, (1d) image quality and (1e) the general appearance of the habitat (cage or enclosure). The design elements that are flexible but impact the compatibility are (2a) camera model, (2b) compute interface for capturing frames, (2c) lighting conditions, (2d) strains and ages of mice, and (2e) animal bedding contents and quantity. Design elements that have no impact on compatibility are (3a) height of habitat walls to prevent mice from escaping, (3b) animal husbandry concerns, (3c) mounting hardware, (3d) technician ergonomic considerations and (3e) electrical connection hardware and management.

### Group 1 specifications

Generally speaking, multi-view imaging will provide the most accurate pose models but requires increased resources on both hardware setup as well as processing of data. Our system operates on a top-down camera viewpoint. This specification enables flexibility and allows more diverse downstream hardware and ease of construction. Top-down provides the advantage of flexibility for materials, since the floor does nott need to be transparent. Additionally, lighting and potential reflection with the bottom-up perspective. Since the paws are not occluded from the bottom-up perspective, models should have improved paw keypoint precision allowing the model to observe more subtle behaviors. However, the appearance of the arena floor will change over time as the mice defecate and urinate. Care must be taken to clean the arena between recordings to ensure transparency is maintained. This doesn't impact top-down imaging that much but will occlude or distort from the bottom-up perspective. Additionally, the inclusion of bedding for longer recordings, which is required by IACUC, will essentially render bottom-up imaging useless because the bedding will completely obscure the mouse.

Our algorithms are trained using data originating from 800 x 800 pixel resolution image data and 30 frames per second temporal resolution. This resolution was selected to strike a balance between resolution of the data and size of data produced. While imaging at higher spatial and temporal

resolution is possible and sometimes necessary for certain behaviors, these values were selected for general mouse behavior such as grooming, gait, posture, and social interactions. Generally, there are trade-offs between frame rate, resolution, color/monochrome, and compression. Some labs have collected data at hundreds of frames per second to capture the kinetics of reflexive behavior for pain (*Jones et al., 2020*) or whisking behavior. Other labs have also collected data at a low 2.5 frames per second for tracking activity or centroid tracking. The data collection specifications are largely dependent on the behaviors being captured. Our rule of thumb is the Nyquist Limit, which states that the data capture rate needs to be twice that of the frequency of the event. For example, certain syntaxes of grooming occur at 7 Hz and we need 14 FPS to capture this data.

We train and test our developed algorithms against the spatial resolution. We note that these are minimum requirements, and down-sampling higher resolution and frame rate data still allows our algorithms to be applied.

Similar to the pixel resolution, we also specify the field of view and imaging distance for the acquired images in real-world coordinates. These are necessary to achieve similar camera perspectives on imaged mice. Cameras must be mounted at a working distance of approximately 100 cm above the floor of the arena. Additionally, the field of view of the arena should allow for between 5 and 15% of the pixels to view the walls (field of view between 55 cm and 60 cm). Having the camera a far distance away from the arena floor reduces the effect of both perspective distortion and barrel distortion. We selected values such that our custom camera calibrations are not necessary, as any error introduced by these distortions is typically less than 1%.

Additionally, image quality is important for meeting valid criteria for enabling the use of machine learning algorithms. Carefully adjusting a variety of parameters of hardware and software values in order to achieve similar sharpness and overall quality of the image is important. While we cannot provide an exact number or metric to meet this quality, users of our algorithms should strive for equal or better quality that exists within our training data. One of the most overlooked aspects of image quality in behavioral recordings is image compression. We recommend against using typical software-default video compression algorithms and instead recommend using either defaults outlined in the software we use or recording uncompressed video data. Using software defaults will introduce compression artifacts into the video and will affect algorithm performance. For example, most video recording software will default to having a constant keyframe rate, typically around every 30 frames. This will artificially create a measurable signal in video features at the keyframe rate.

Finally, the general appearance of the cage should be visually similar to the variety of training data used in training the machine learning algorithms. Documentation on this for each individual algorithm for assessing the limitations is published (*Sheppard et al., 2022*; *Geuther et al., 2021*; *Hession et al., 2021*; *Geuther et al., 2019*). While our group strives for the most general visual diversities in mice behavioral assays, we still need to acknowledge that any machine learning algorithms should always be validated on new datasets that they are applied to. Generally, our machine learning algorithms earlier in the entire processing pipeline, such as pose estimation, are trained on more diverse datasets than algorithms later in the pipeline, such as pain and frailty predictions.

## Group 2 specifications

In order to achieve compliant imaging data for use with our machine learning algorithms, we specify the hardware we use. While the hardware and software mentioned in this section is modifiable, we recommend that careful consideration is taken such that changes still produce compliant video data.

We modified a standard open field arena that has been used for high-throughput behavioral screens (*Kumar et al., 2011*). The animal environment floor is 52 cm square with 92 cm high walls to prevent animals escaping and to limit environmental effects. The floor was cut from a 6 mm sheet of Celtec (Scranton, PA) Expanded PVC Sheet, Celtec 700, White, Satin / Smooth, Digital Print Gradesquare and the walls from 6 mm thick Celtec Expanded PVC Sheet, Celtec 700, Gray, (6 mm x 48 in x 96 in), Satin / Smooth, Digital Print Grade. All non-moving seams were bonded with adhesive from the same manufacturer. We used a Basler (Highland, IL) acA1300-75gm camera with a Tamron (Commack, NY) 12VM412ASIR 1/2" 4–12 mm F/1.2 Infrared Manual C-Mount Lens. Additionally, to control for lighting conditions, we mounted a Hoya (Wattana, Bangkok) IR-80 (800 nm), 50.8 mm Sq., 2.5 mm Thick, Colored Glass Longpass Filter in front of the lens using a 3D printed mount. Our cameras are mounted 105+/-5 cm above the habitat floor and powered the camera using the

power over ethernet (PoE) option with a TRENDnet (Torrance, CA) Gigabit Power Over Ethernet Plus Injector. For IR lighting, we used 6 10-inch segments of LED infrared light strips (LightingWill DC12V SMD5050 300LEDs IR InfraRed 940 nm Tri-chip White PCB Flexible LED Strips 60LEDs 14.4 W Per Meter) mounted on 16-inch plastic around the camera. We used a 940 nm LED after testing an 850 nm LED which produced a marked red hue. The light sections were coupled with the manufactured connectors and powered from a 120VAC:12VDC power adapter.

For image capture, we connected the camera to an nVidia (Santa Clara, CA) Jetson AGX Xavier development kit embedded computer. To store the images, we connected a local four-terabyte (4TB) USB connected hard drive (Toshiba [Tokyo, Japan] Canvio Basics 4TB Portable External Hard Drive USB 3.0) to the embedded device. When writing compressed videos to disk, our software both applies optimized de-noising filters as well as selecting low compression settings for the codec. While most other systems rely on the codec for compression, we rely on applying more specific de-noising to remove unwanted information instead of risking visual artifacts in important areas of the image. We utilize the free ffmpeg library for handling this filtering and compression steps with the specific settings available in our shared C++ recording software. Complete parts list and assembly steps are described in (https://github.com/KumarLabJax/JABS-data-pipeline).

## Group 3 specifications

Finally, here we present hardware and software that can be modified without risk of affecting video compliance. For natural light, we used a F&V (Netherlands) fully dimmable R-300SE Daylight LED ring light powered by a 120vac:12vdc power adapter. These lights are adjustable to meet the visible lighting needs of specific assays without affecting the visual appearance of the data. To keep the animals nourished, we installed water bottles and a food hopper external to the animal environment. These were placed on the outside of the arena on a removable panel. The panel can be customized as needed for experiments without the need to replace/modify the entire arena. To suspend the camera and lights, we used a wire shelf from our solution for technician ergonomics.

To raise the animal cage to an ergonomic height, we used the 24-inch by 24-inch option of the Metro (Wilkes-Barre, PA) Super Erecta wire shelving system with three shelves. As mentioned in the earlier paragraph, the topmost shelf was used to suspend the camera and lights. We also hinged one wall, turning it into a door, to allow easier animal access. Communication between the electronic devices was interconnected with CAT5 cables and a network switch, and a powered USB hub was used between the USB connected hard drive and the nVidia compute device. We used a digital timer for the visible LED light, a 120 v power strip to consolidate the power, and a universal power source (battery backup) between the chamber and facility power.

For ease of use and reduction of environmental noise, we also include a software for remote monitoring and welfare check. The software consists of three main components: a recording client implemented in C++, a control server implemented with the Flask Python framework, and a web-based user interface implemented with Angular (*Figure 1*). The recording client runs locally on each Nvidia Jetson Xavier computer and communicates with the server using the Microsoft C++ REST SDK to provide centralized monitoring and control of distributed recording devices. The recording client captures raw frames from the camera and encodes video using the ffmpeg library. In addition to saving encoded video on the local hard drive, the recording client can optionally send video over the RTMP protocol to an NGINX server configured with the nginx-rtmp plug-in. The web interface communicates with the control server, which relays recording start and stop commands to individual recording devices, enabling the user to remotely control various aspects of recording in addition to viewing the live stream from the NGINX streaming server using the HTTP Live Streaming (HLS) protocol (*Figure 2*). Each JABS-DA unit has its own edge device (Nvidia Jetson). Each system (which we define as multiple JABS-DA areas associated with one lab/group) can have multiple recording devices (arenas). The system requires only 1 control portal (RPi computer) and can handle as many recording devices as needed (Nvidia computer w/ camera associated with each JABS-DA arena). To collect data, 1 additional computer is needed to visit the web control portal and initiate a recording session. Since this is a web portal, users can use any computer or a tablet. The recording devices are not strictly synchronized but can be controlled in a unified manner.

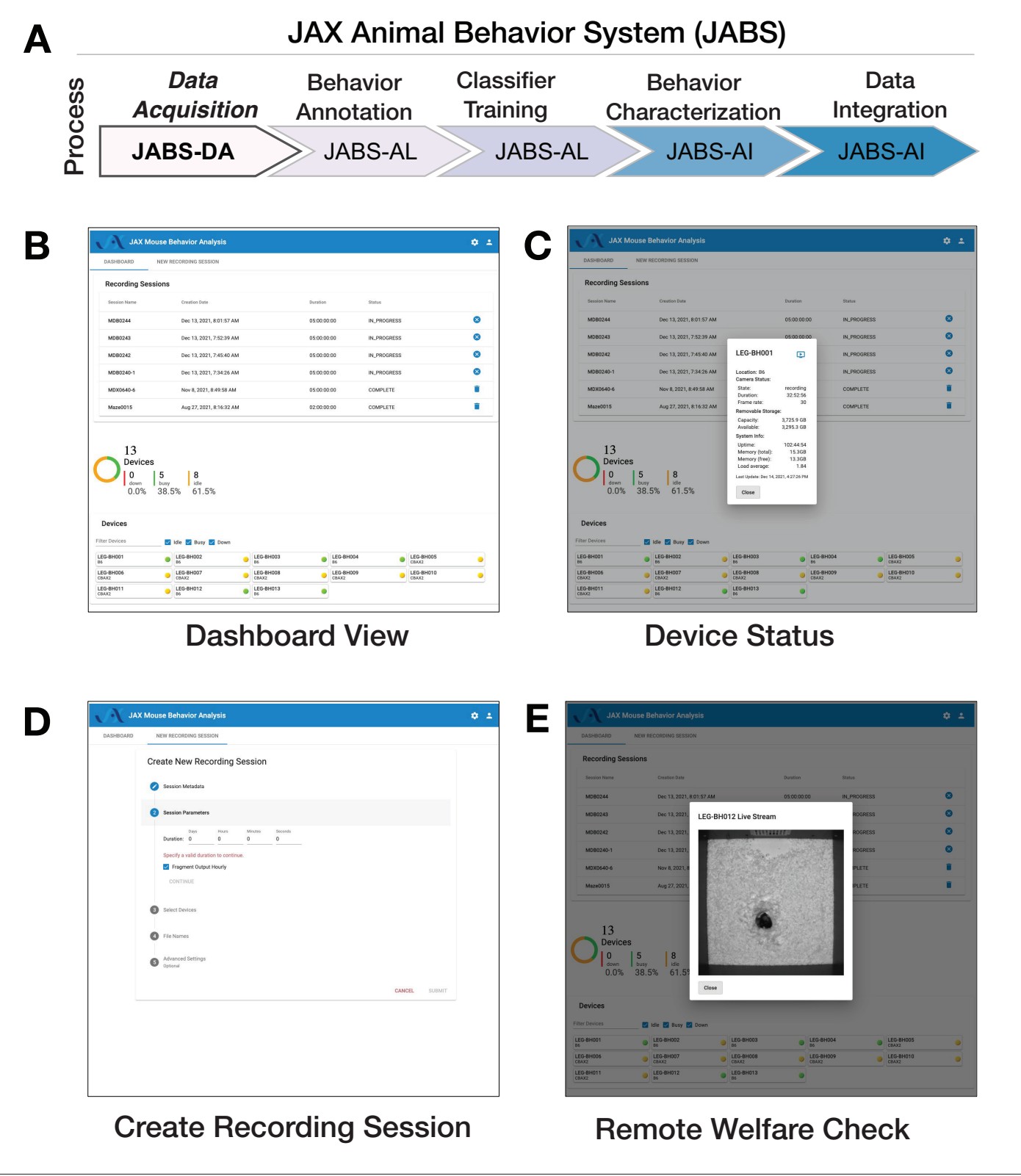

**Figure 2.** JABS data acquisition module (JABS-DA) consists of a web-based control system for recording and monitoring experiments. (**A**) JABS pipeline. (**B–E**) Screenshots from Angular web client that allows monitoring of multiple JABS Acquisition units in multiple physical locations can be seen on one screen (**B**). Dashboard view allows monitoring of all JABS units and their status, Device Status provided detailed data on individual devices (**C**) Recording session dashboard allows initiation of new experiments (**D**), and remote welfare view allows live video to be streamed from each unit (**E**).

*Figure 2 continued on next page*

*Figure 2 continued*

The online version of this article includes the following figure supplement(s) for figure 2:

**Figure supplement 1.** JABS data acquisition module: Environmental parameters in the arena.

**Figure supplement 2.** Representative hematoxylin and eosin (H&E) stained tissue sections from mice after spending 14 days in the JABS arena or control wean cage.

## Environment checks

To evaluate the suitability of JABS-DA for long-term housing of mice, we conducted a series of experiments comparing environmental conditions and animal health outcomes within these arenas to those observed in standard JAX housing cages. Our goal was to provide data for the JAX Institutional Animal Care and Use Committee (IACUC) to confirm health and welfare of animals over time in these apparatus. These data can be used for Institutional ACUC protocols by others. We compare our data with established guidelines from the Guide for the Care and Use of Laboratory Animals (the Guide; *National Research Council, 2011*). Our experiments were performed in one room at The Jackson Laboratory, Bar Harbor, Maine (JAX) with temperature and humidity set to 70–74 °F (~21–23 °C) and 40–50%, respectively.

One concern related to the use of the JABS arena in long-term experiments was that the 90 cm height of the walls without lower air openings might result in inadequate air flow and build-up of toxic gases. To address this, we compared environmental parameters in JABS arenas with that of a standard JAX housing cage. Two JABS arenas were observed with 12 male C57BL/6 J mice 12–16 weeks old in each for a 14-day period. At the same time, one wean cage containing 10 male C57BL/6 J age-matched mice was observed on a conventional rack for matching air flow in the same room. We used a #2 Wean Cage (30.80 x 30.80 x 14.29 cm) from Thoren (Hazleton, Pennsylvania) with 727.8 cm$^2$ floor space, which is a common housing container for mice and is approved at JAX to house 10 animals. This commercial cage has a floor area that is ~1/4 that of the JABS arena. The ceiling height in the wean cage ranges from 5–14 cm due to the sloped metal wire cover that contains food and water. The JABS arena, by contrast, has no ceiling. Food, water, bedding type and depth, and light level were all matched in the arenas and wean cage. Bedding (1:1 ratio of aspen chip/shavings mix and Alpha-Dri) was left unchanged for the full 2-week period to minimize interaction with mice in JABS arenas as much as possible. To determine if forced air flow was needed for an acceptable arena environment, one of the two arenas and the wean cage were exposed to normal room air flow, while the second arena had a 6-inch quiet electric fan mounted above for increased circulation. The fan was pointed to blow air up to draw air out of the arena instead of actively blowing air towards the mice.

We monitored $CO_2$ and ammonia, common housing gases (*National Research Council, 2011*). $CO_2$ was measured with an Amprobe $CO_2$ meter daily, excluding weekends and holidays, in both arenas and the wean cage. $CO_2$ was recorded in the room's center before and after each arena and wean cage measurement as a control. For higher levels, $CO_2$ is shown as a range due to oscillation. Ammonia was tested with Sensidyne Ammonia Gas Detector Tubes (5–260 ppm) in the arena without a fan and the wean cage on days 0, 2, 4, 7, and 14, with samples taken near the floor and waste accumulation areas. Temperature and humidity data loggers (MadgeTech RHTEMP1000IS) were placed on the floor in each arena and the wean cage for the experiment's duration. An environment monitor (Hobo, U12-012, Onset) was mounted on the wall for room background data. Body weight was measured daily, excluding weekends and holidays. Grain and water were weighed at the start and end of each experiment to check consumption.

We observed daily room background $CO_2$ levels of 454–502 ppm throughout the 14-day experiment. These are very close to expected outdoors levels and indicative of a high air exchange rate (*Myatt et al., 2002*). JABS arena $CO_2$ levels varied from a low of 511 ppm on day 1 to an oscillating range of 630–1565 ppm on day 14. The JAX standard wean cage experienced an oscillating range of 2320–2830 ppm on day 0, climbing to an oscillating range of 3650–4,370 ppm on day 14. The wean cage $CO_2$ values approximately match those from another published study of maximum grouped mice in similar static housing (*Mexas et al., 2015*). Indoor $CO_2$ is often evaluated as a level above background (*Myatt et al., 2002*). We observe a maximum JABS arena $CO_2$ level above background of 1082 ppm. This is 3.8-fold lower than the maximum observed $CO_2$ levels in the wean cage (4121 ppm; *Figure 2—figure supplement 1B*, arena with fan excluded from graph for clarity).

Ammonia levels in the JABS arena were below 5 ppm on days 0, 2, 4, and 7, rising to 18 ppm on day 14. In the wean cage, levels were <5 ppm on days 0 and 2, rose to 52 ppm on day 4, and remained at ~230 ppm on days 7 and 14. Initial concerns about high JABS arena walls hindering airflow were alleviated as $CO_2$ and ammonia levels indicated better air exchange than standard housing. NIOSH's recommended maximum ammonia exposure for humans is 25 ppm over 8 hours, with a similar recommendation for mice (*National Research Council, 2011*; *Fawcett and Rose, 2012*). Ammonia levels are mainly influenced by air changes per hour (ACH) (*Gamble and Clough, 1976*; *Ferrecchia et al., 2014*). JAX animal rooms have ~10 ACH and PIV cages have ~55–65 ACH. Ammonia levels were consistently 10–50 times lower in the JABS arena compared to the control static wean cage and remained well within recommended limits (*Figure 2—figure supplement 1C*). Future JABS arena observations must consider the impact of ammonia on behavior (*Tepper et al., 1985*). Mice used in JABS experiments come from PIV housing, where ammonia levels are expected to be similar to those in the JABS arena, minimizing behavioral impact (*Ferrecchia et al., 2014*).

Temperatures in all locations (room background, two JABS arenas, and one wean cage) remained in a range of 22–26 °C throughout the experiment. Variance in room background readings suggests temperature fluctuations are more due to innate room conditions (such as environmental controls) than anything else. We find that arena structure does not adversely affect control of the temperature to which mice are exposed (*Figure 2—figure supplement 1D*).

The probes that measured temperature also measured humidity. The room probe, mounted on a wall 1 foot above the floor in the 8x8 feet room, recorded consistent background humidity of 45 ±5% (*Figure 2—figure supplement 1E*, green line). Housing probes in the bedding of each chamber—centered in JABS arenas and along a wall in the smaller wean cage—recorded 55–60% humidity in the JABS arenas, except for occasional spikes not correlated with background changes, likely due to mouse urination (*Figure 2—figure supplement 1E*, blue and black lines). In contrast, wean cage humidity rose from 55–60% to above 75% within 12 hr and continued climbing to 97.5% by day 14 (*Figure 2—figure supplement 1E*, red line). Higher humidity in the microenvironments was due to mouse urination and limited air flow (Guide *National Research Council, 2011*). The JABS arenas maintained a drier environment because they had a higher bedding to mouse ratio (3.2 times more per mouse) and better air circulation compared to the wean cage (*Figure 2—figure supplement 1E*).

Weight is often used as a marker for health, although body condition score is used as a more reliable indicator of serious concerns (*Easterly et al., 2001*; *Hickman and Swan, 2010*; *Guzman et al., 2023*). Mice in JABS arenas lost weight compared to those in the wean cage, and this was initially a cause of concern. However, mice in JABS arenas maintained a healthy appearance and normal body condition score throughout the experiment. Other measurements demonstrating normal parameters and other control experiments not shown additionally led us to believe the weight differences are because JABS arena mice are active while wean cage mice, with more limited movement available, are sedentary. Mice started the experiment at 25–33 g body weight. The lowest average recorded during the experiment was 95.6% of the start value, for mice in the JABS arena without a fan on day 9. The lowest individual recorded was 85.8% of start value at 23.6 g on day 14, also in the arena without a fan (*Figure 2—figure supplement 1F*).

Per mouse grain usage was comparable between the JABS arena and the wean cage and in an expected range (*Lovasz et al., 2020*; *Figure 2—figure supplement 1G*). Per mouse water usage was comparable between the JABS arena and the wean cage and in the expected range (*Green, 1966*). Somewhat higher water use in the arena could be indicative of higher activity requiring more hydration (*Figure 2—figure supplement 1H*). Since only one JABS arena and one wean cage were tested, error bars are not available to aid in interpretation.

Three mice from one arena and three from a wean cage were necropsied immediately following 14 days in the JABS arena or control wean cage to determine if any environmental conditions, such as possible low air flow in arenas potentially leading to a buildup of heavy 'unhealthy' gases like ammonia or $CO_2$, were detrimental to mouse health. Nasal cavity, eyes, trachea, and lungs were collected from each mouse. They were H&E stained and analyzed by a qualified pathologist. No signs of pathology were observed in any of the tissue samples collected (*Figure 2—figure supplement 2*).

Based on these environmental and histological analyses, we conclude that the JABS arena is comparable and in many respects better than a standard wean cage. Lack of holes near the floor does not create a buildup of ammonia or $CO_2$. Mice ate and drank at normal levels. We initially observed a

slight decrease in body weight, which increased in the next few days. We hypothesize that this could be due to the novel environment and the increase in space for movement, leading to more active mice.

## JABS-AL: an active learning module for behavior classifier training

In this section, we first present an overview of behavior annotation and classifier training using the JABS-AL module, which utilizes our python-based, open-source graphical user interface (GUI) application that has been developed to be compatible with Mac, Linux, and Windows operating systems. We then evaluate the utility and accuracy of JABS trained classifiers through two complementary approaches. In the first approach, we benchmark the performance of JABS classifiers against a previous neural network-based approach (*Geuther et al., 2021*), providing us a comparison of the performance of the two approaches on the same dataset. In the second approach, we studied how classifiers for the same behavior trained by two different human annotators in the lab compare with each other in terms of behavior identification, allowing us to assess the inherent variability among expert annotators.

### Behavior annotation and classifier training

There are two prominent approaches in the literature for training behavioral classifiers. The first approach trains the classifiers using the raw video files, as previously demonstrated to identify grooming behavior through the use of a deep neural network (*Geuther et al., 2021*; *Bohnslav et al., 2021*). The second approach involves first extracting pose keypoints in each frame using deep neural networks, which serve as inputs for machine learning classifiers (*Biderman et al., 2024*; *Mathis et al., 2018*; *Pereira et al., 2022*). Previously, we utilized a deep neural network-based classifier to extract poses and used the keypoints to study gait behavior (*Sheppard et al., 2022*). A pose-based approach offers the flexibility to use the identified poses for training classifiers for multiple behaviors, and we used this approach for JABS. The advantage lies not just in training speed, but in the transferability and generalization of the learned representations. Pose-based approaches create structured, low-dimensional latent embeddings that capture behaviorally relevant features which can be readily repurposed across different behavioral classification tasks, whereas pixel-based methods require retraining the entire feature extraction pipeline for each new behavior. Recent work demonstrates that pose-based models achieve greater data efficiency when fine-tuned for new tasks compared to pixel-based transfer learning approaches (*Ye et al., 2024*), and latent behavioral representations can be partitioned into interpretable subspaces that generalize across different experimental contexts (*Whiteway et al., 2021*). While pixel-based approaches can achieve higher accuracy on specific tasks, they suffer from the 'curse of dimensionality' (requiring thousands of pixels vs. 12 pose coordinates per frame) and lack the semantic structure that makes pose-based features inherently reusable for downstream behavioral analysis. Additionally, the extracted keypoints can also be used to generate quantifiable and interpretable features that can be used to study various aspects of animal behavior such as gait and posture. In addition to the raw video file, JABS annotation and classification active learning module requires pose files from our previously established neural network for pose estimation as an input to train the classifiers. Note that the raw videos are needed only for annotating behaviors, and one can predict the behaviors using only the pose files.

We have developed an easy-to-use open-source Python GUI software to annotate behaviors in videos, as shown in *Figure 3A*. This tool allows users to easily annotate behaviors in video recordings through mouse/trackpad or keyboard shortcuts, as well as the option to leave frames unlabeled for ambiguous cases. The GUI provides statistics of the total number of frames as well as the number of frames and bouts annotated for a particular behavior. The annotations are displayed below the video as an ethogram (*Figure 3B*). The user can annotate multiple behaviors for the same video. Once the minimum number of frames (100) and videos (2) have been annotated, the user can train a classifier using either of the tree-based methods such as Random Forest (RF)/Gradient Boost/XGBoost (XGB) (*Chen and Guestrin, 2016*; *Ho, 1995*; *Breiman, 2001*; *Friedman, 2001*) and check the classifier's accuracy with k-fold cross-validation, selecting a value of k that balances computational efficiency and accuracy. The input features for these classifiers are derived from the animal's pose, which must be estimated prior to using the GUI. This is accomplished using our separate pose tracking pipeline (https://github.com/KumarLabJax/mouse-tracking-runtime, *Geuther et al., 2026b*), which employs

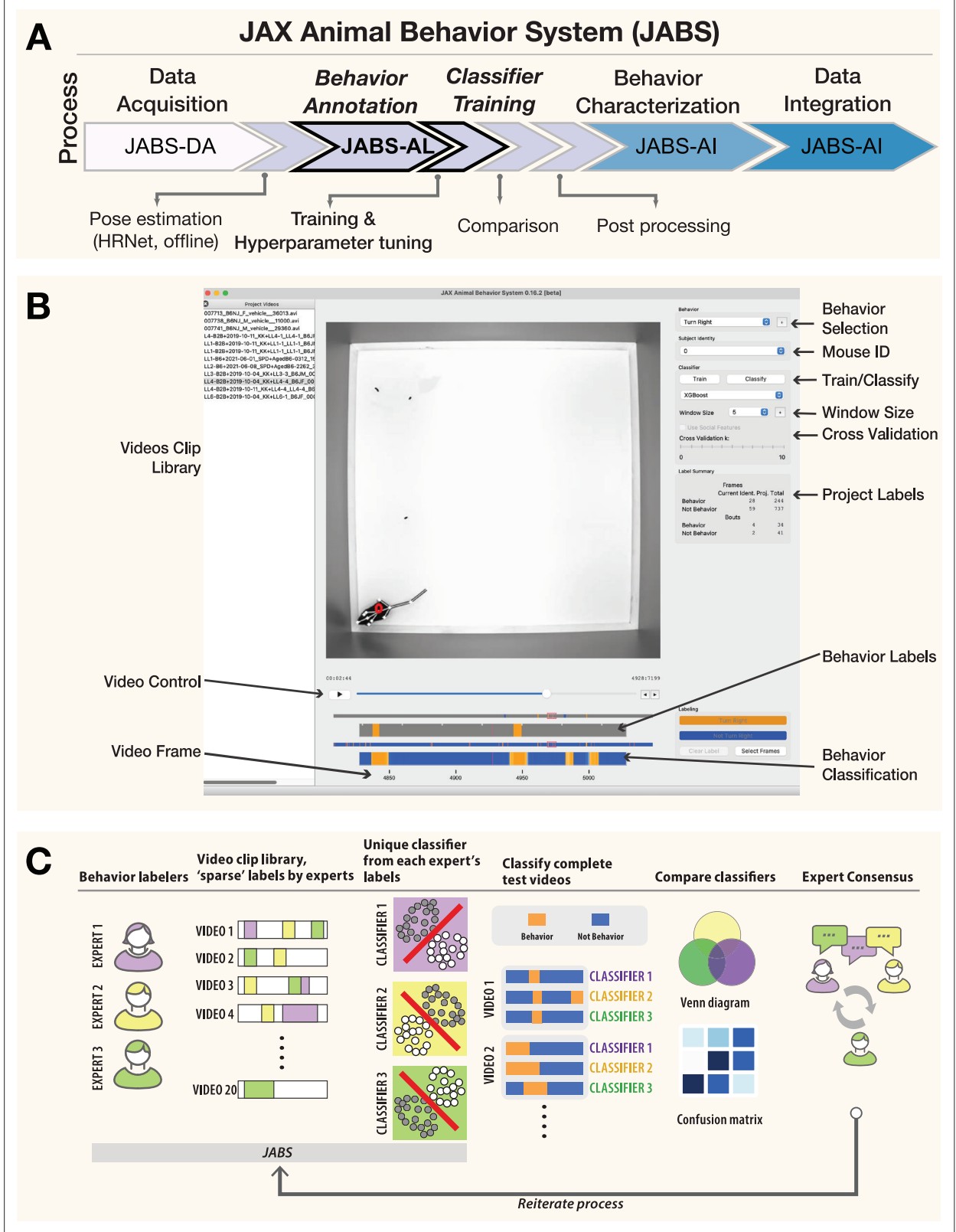

**Figure 3.** JABS-AL is a behavior annotation and classification module that allows training classifiers with sparse labels. (**A**) JABS pipeline highlighting individual steps towards automated behavioral quantification. (**B**) Screenshot of the Python-based open-source GUI application used for annotating multiple videos frame by frame. One can annotate multiple mice and for multiple behaviors. The labeled data is used for training classifiers using either random forest or gradient boosting methods. Adjustable window size (number of frames on the left and right of the current frame) to include features

*Figure 3 continued on next page*

*Figure 3 continued*

from a window of frames around the current frame. The labels and predicted labels are displayed at the bottom. (**C**) A sample workflow for training a typical classifier. Multiple experts can sparsely label videos to train multiple classifiers for the same behavior. These classifiers can be compared, and experts can consult to iterate through the training process.

an HRNet-based neural network (*Sheppard et al., 2022*) to identify the location of twelve keypoints in each video frame. This pipeline is described in greater detail in the Materials and methods section. We then compute a number of informative features like distance between various keypoints, linear and angular velocity between keypoints, etc. that are used as input for these classifiers. We also incorporate temporal information from the videos by computing window features that include information from $w$ (window size) frames on each side of the current frame. A complete list of base features currently included in JABS is provided in the *Supplementary file 3*. The weights of different features used by the trained classifiers improve the interpretability of the classifiers.

Typically, to arrive at an optimal classifier for a behavior, we start by training multiple classifiers using annotated data from different human experts for the same set of videos and then evaluating the performance of each classifier against a separate set of test videos as depicted in *Figure 3C*. However, a core challenge is that different researchers may 'see' and define the same behavior differently. This leads to low inter-rater reliability, where two people annotating the same video produce different labels. Disagreements often arise from two primary sources: the granularity of the labels and their reliance on subjective interpretation versus objective, physical descriptions. These are often referred to as labeling style due to annotator encoding their intuition about the behavior (*Kabra et al., 2013*; *Segalin et al., 2021*). Even when there is high interannotator agreement, explainable machine learning methods have shown that labelers rely on different features for labeling the same behavior (*Goodwin et al., 2024*). Added frustration results when experts have to redefine and relabel videos to increase interannotator agreement. Researchers often try to break down complex behaviors into heuristic rules for the labelers to follow, which almost always leads to conflict between these rules and intuitive understanding of the complex behavior. These predetermined rules never capture the full repertoire of a complex behavior and often ask the labeler to violate their intuition.

In JABS, we advocate that each expert sparsely labels frames from a video library to create a behavior classifier. These are essentially expert-specific classifiers that encode the intuition or labeling style of that behaviorist. These classifiers are inferred on a set of test videos that have been set aside (*Figure 3C*). Each expert's classifiers are inferred from these test videos. We can then compare interannotator agreement using these inferred sense labeled videos. Experts can confer among themselves to reach a consensus on whether they agree on the inference of each classifier, potentially altering their definition of the behavior and altering their labeling style (*Figure 3C*). JABS allows the experts to reiterate through the labeling process to arrive at a consensus. The process gives expert labelers an opportunity to agree on a behavior definition but does not operationalize the behavior through rules. In the end, there may be multiple expert-specific classifiers for the same behavior, and end users must decide which classifiers they prefer to use. In this paper, we propose the use of broad-sense heritability to prioritize classifiers for the same behavior derived from different experts' labels.

After classifiers are finalized, we often ask experts to densely label (label every frame) another small set of test videos, which are used for final validation. We find that creating the dense label dataset at the end allows the labelers to have a clear understanding and intuitive definition of the behavior, whereas creating the dense label in the beginning of a project leads to frequent shifts in behavior definition as the expert sees more instances of a behavior, particularly edge cases. The detailed user guide, along with a video tutorial on how to install and run the JABS Active Learning app, is available online (here).

## Benchmarking JABS classifier using grooming behavior

Previously, a CNN-based grooming behavior classifier trained on raw videos attained human-level accuracy (*Geuther et al., 2021*). We re-purpose this large training dataset as a benchmark for estimating learning capacity of pose-based classifiers. For context, we report JABS classifier performance against both JAABA and CNN in *Geuther et al., 2021*. Further, we evaluate how the performance of the classifier varies with the choice of machine learning algorithm, window size ($w$) of the features and the amount of training data. For the choice of machine learning algorithm, we utilize two popular

tree-based methods, namely Random Forest (RF) and XGBoost (XGB). Briefly, the dataset contains 1253 video segments, and we held out 153 video clips for validation (this is the same validation set used in *Geuther et al., 2021*) and the rest are used to train the classifier. This split results in similar distributions of frame-level classifications between training and validation sets. More details of the dataset are available in *Supplementary file 1*. We trained multiple classifiers by varying the amount of annotated data, window size, and machine learning algorithm. Our best accuracy from the neural network-based approach for this dataset was 0.937, and the best classifier from JABS using all the annotated data, a window size ($w$) of 60 frames, and XGB machine learning algorithm achieved a comparable per-frame accuracy score of 0.9364. We noticed that with the same set of features, XGB typically achieved better accuracy than the RF method across different window sizes and training data size. The results for these benchmark tests are shown in *Figure 4B–D*. Our tests with different window sizes show that grooming performance increases as we increase the window size, reaches a maximum (around 60 frames), and then degrades for large window sizes (*Figure 4B*). Because grooming typically lasts for a few seconds, classifiers using features within nearby frames will perform better as they incorporate optimal temporal information, and including features from too few or too many frames will decrease the performance. We also investigate the impact of the amount of labeled data on the performance of JABS classifiers, as it can help to optimize the annotation process, ultimately reducing the time and resources required to train the model. To do this, we trained the XGB and RF classifiers using a subset of the full dataset (about 20 hr) consisting of 10, 20, 50, 100, 500, and 1100 training videos. These correspond to approximately 1.3%, 2.2%, 4.4%, 8.5%, 46.1%, and 100% out of a total of 2,181,790 frames. As expected, the performance of JABS improves as we include more labeled data. However, the results demonstrate that a high degree of accuracy, approaching 85%, can be attained through the utilization of only 10 videos of training data, as evidenced by the corresponding area under the receiver operating characteristic curve (AUROC) of approximately 0.94, as depicted in *Figure 4C–E*.

Additionally, it was found that the true positive rate (TPR) experienced a minimal decrease of about 1% when the training data was reduced from 100% to 50%, while maintaining a false positive rate (FPR) of 5% (*Figure 4F*). To assess the generalizability of our grooming classifier across genetic diversity, we trained the model on videos spanning 60 genetically diverse mouse strains (n=1100 videos) and evaluated performance on an independent test set comprising 51 genetically diverse strains (n=153 videos). Per-strain analysis revealed robust and uniform performance across the majority of genetic backgrounds (median F1=0.94, IQR = 0.8990.956). We observed only modest performance declines in albino strains, attributed to reduced visual contrast under infrared illumination conditions (see *Figure 4—figure supplements 1 and 2*).

In the rapidly evolving field of automated quantification of animal behavior, two predominant methodologies have been established for learning behavior: using raw video data and using a reduced representation (abstraction) of the animal with certain keypoints, from which informed features are calculated (*Sheppard et al., 2022*; *Segalin et al., 2021*; *Mathis et al., 2018*; *Kabra et al., 2013*; *Goodwin et al., 2024*). To understand the trade-offs and strengths of each approach, we evaluate the performance of different classifiers that employ these methodologies when utilizing varying amounts of training data, as depicted in *Figure 4G*. Interestingly, our findings demonstrate that utilizing keypoint-based low dimensional representation of animal behavior, as employed by JABS and JAABA (*Kabra et al., 2013*) methodologies, leads to superior performance when compared to using high dimensional raw video data as employed by 3D CNNs, particularly when the availability of training data is limited. However, as the quantity of training data increases, the performance of both approaches tends to converge.

Therefore, by distilling the essence of a video into a series of key poses, JABS is able to effectively learn and generalize, even with smaller training sets. It has been shown to have a learning capacity on par with deep neural networks, as demonstrated by per-frame accuracy using the same benchmark data set. Further, achieving 85% accuracy with just 1.4% of the labeled data suggests that researchers can strike a balance between labeling efforts and desired accuracy by carefully selecting the amount of training data.

## JABS analysis and integration module

In supervised machine learning, the accuracy and reliability of a trained classifier depends heavily on the quality of labeled data. Further, it has been observed that labeling of the same behavior by

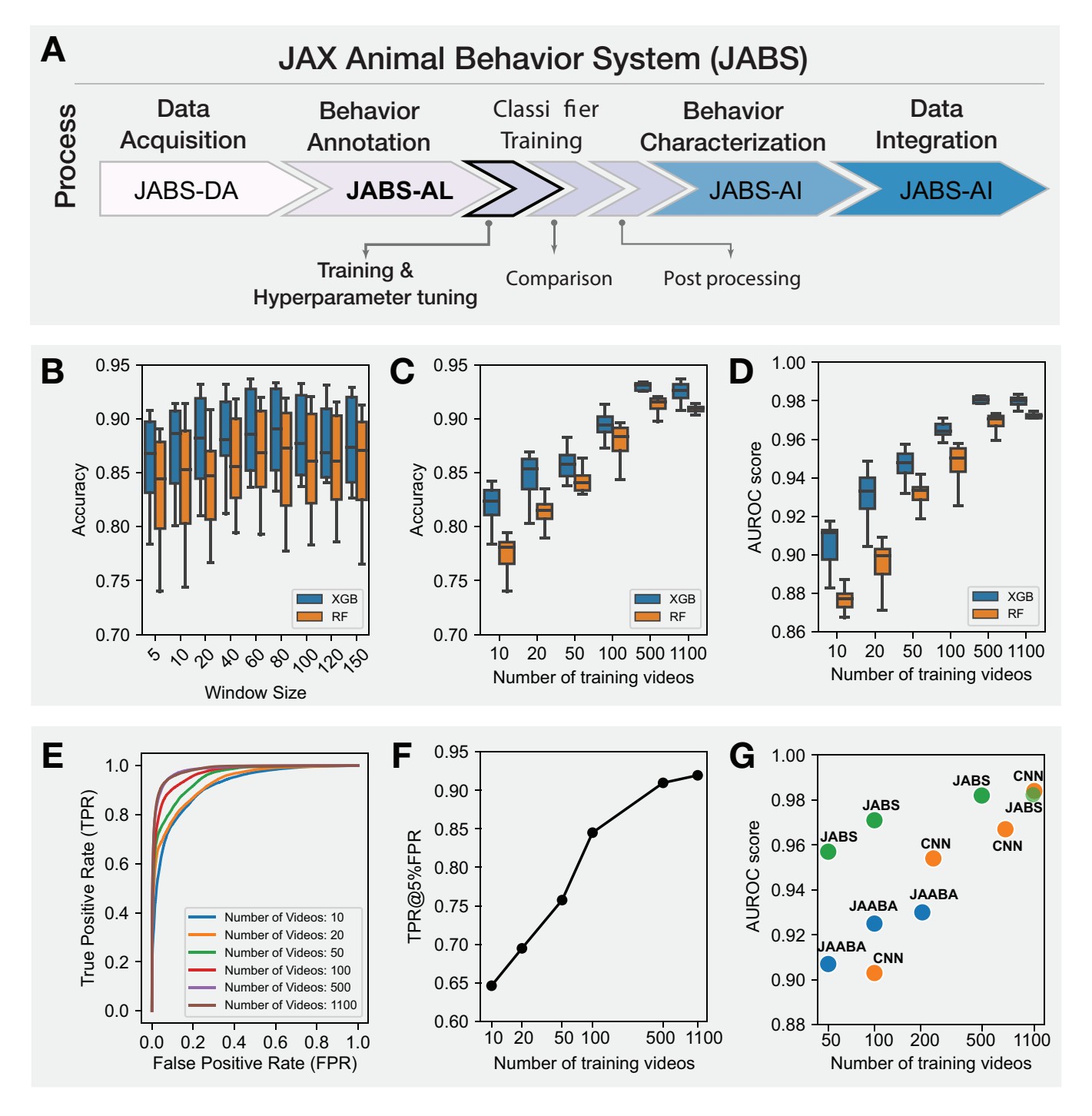

**Figure 4.** JABS Benchmarks: Selecting hyper-parameters and benchmarking JABS classifiers using grooming dataset. (**A**) JABS pipeline highlighting individual steps towards automated behavioral quantification. Using feature window size, type of classification algorithm, and the number of training videos as our benchmarking parameters: (**B**) Accuracy of JABS classifiers trained using different window size (in frames) features. Each boxplot shows the range of accuracy values for different number of training videos and type of classification algorithms. (**C, D**) The effect of increasing the training data size on Accuracy and AUROC score of the JABS classifiers. (**E**) ROC curves for the JABS classifier trained with the window size of 60, XGB algorithm and varying training data size. (**F**) True positive rate at 5% false positive rate corresponding to the JABS classifier from panel (**E**) as the amount of training data is changed. (**G**) Comparing the performance of JABS-based classifiers with a 3D Convolutional neural network (CNN) and JAABA-based classifiers for different training data sizes. JAABA and CNN results were adopted from *Geuther et al., 2021*.

*Figure 4 continued on next page*

*Figure 4 continued*

The online version of this article includes the following figure supplement(s) for figure 4:

**Figure supplement 1.** Per strain F1 score and accuracy for grooming dataset.

**Figure supplement 2.** F1 score and accuracy for mice with different coat colors in the grooming dataset.

different human experts introduces variability among annotations due to a variety of factors, including personal biases, subjectivity, and individual differences in understanding what constitutes a behavior (*Kaufman and Rosenthal, 2009*; *Tjandrasuwita et al., 2021*). Therefore, it is critical to accurately capture the inter-annotator variability before selecting classifiers for downstream predictions. To capture this variability, we employ both frame-based and bout-based comparison and demonstrate that bout-based comparison gives a better estimate of inter-annotator agreement.

## Frame and bout-wise classifier comparison of inter-annotator variability

In order to test inter-annotator variability, we generated a set of single mouse behavior classifiers for two simple behaviors, left and right turn. We inferred behavior from all four classifiers on a large set of videos and compared the two pairs of classifiers from each annotator (*Figures 5 and 6*). The classifiers for all behaviors achieved good accuracy and F1 scores can be found in *Supplementary file 1*. Further, the classifiers for the same behavior trained with different human annotations resulted in inter-annotator variability in predictions. This inter-annotator variability can be associated with (a) subjective differences of behavior definition among human labelers (b) varying level of annotator expertise, and (c) training within and across labs. We investigated the source of this variability and sought to determine the best method to mediate its effects. To capture this effect, we first visualized the predictions made by two classifiers trained for the same behaviors (left and right turn) but with different human annotators: annotator-1 (A1) and annotator-2 (A2). *Figure 5B and C* shows two sample ethograms corresponding to the predictions made by A1 and A2 for the left turn behavior. These ethograms show high levels of concordance between the two annotators. However, upon closer examination, we observed that the percentage of left or right turn behavior predicted (for all the videos) by A2 was higher than A1 (see *Figure 5D and G*). The confusion matrix (shown in *Figure 5E and H*) quantifies the level of agreement between predictions made by annotators A1 and A2 for left and right turn behavior. However, since this behavioral task is heavily class-imbalanced (the number of frames with no behavior is much more than that of behavior), accuracy can be misleading, as the classifier can achieve high accuracy by simply predicting majority class (not behavior) for all the frames. To address this imbalance, we calculate Cohen's kappa ($\kappa$) metric (*McHugh, 2012*) which is a commonly used measure of inter-annotator agreement accounting for the class imbalance. Mathematically, it is defined as $\kappa = \frac{p_o - p_e}{1 - p_e}$, where $p_o$ is the observed agreement between annotators and $p_e$ is the expected agreement due to random chance. A $\kappa$ score of 0 indicates that the agreement is no better than chance, and a score of 1 indicates perfect agreement, regardless of high/low accuracy. Finally, we visualize the frame-wise comparison of the two annotators showcasing the percentage of frames where the annotators agree and disagree on the occurrence of a behavior as shown in *Figure 5F and I*. The Venn diagram clearly highlights the discrepancy between high accuracy resulting from class imbalance (*Figure 5E and H*) and significant mismatch between % of predicted behavior (*Figure 5D and G*), with annotator A2 accounting for the majority of discrepancy by predicting more frames as turning behavior compared to annotator A1.

We observed in the ethogram (*Figure 5B and C*) that although many of the same bouts are captured by both A1 and A2, most of the frame discrepancies seem to be in the beginnings and ends of the bout. A2 seems to predict longer bouts than A1 (*Figure 5D*). Between two humans labeling the same behavior, there are unavoidable and sometimes substantial discrepancies in the exact frames of the behavior labeled even when trained in the same lab (*Segalin et al., 2021*; *Tjandrasuwita et al., 2021*). To most behaviorists, detecting the same bouts of behavior is more important than the exact starting and ending frame of these bouts, as again, there are human-level discrepancies in this as well. Therefore, we used a bout-based comparison rather than a frame-based comparison to evaluate the performance of the classifiers.

For the bout-based comparison, we looked at how much overlap there was between the bouts of a behavior predicted by annotators A1 and A2, taking inspiration from the machine learning

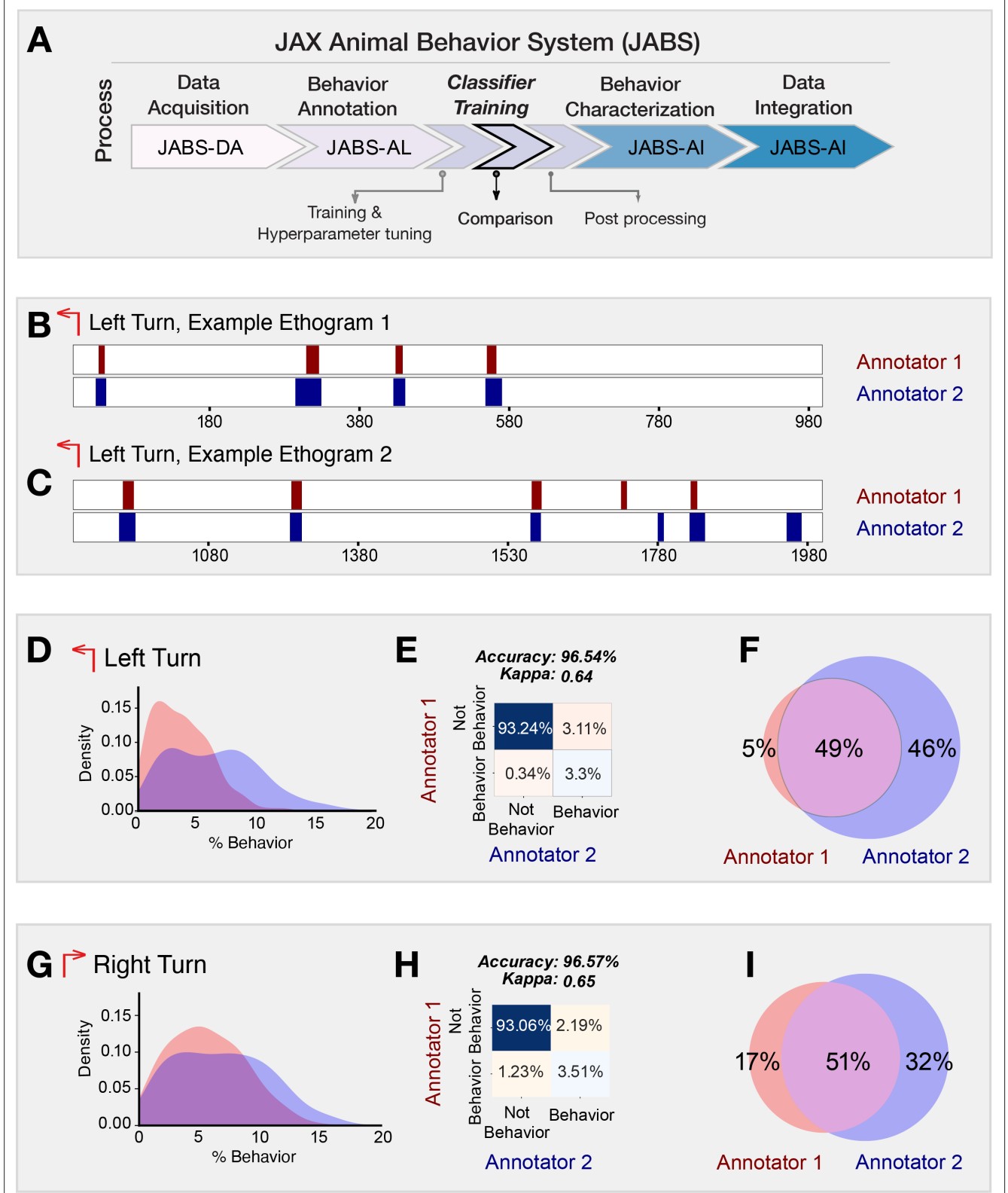

**Figure 5.** Frame-based comparison of classifiers from different annotators but trained for the same behavior. (**A**) JABS pipeline highlighting individual steps toward automated behavioral quantification. (**B, C**) Two sample ethograms for the left turn behavior showing variation in behavior inference for two different annotators. (**D, G**) Kernel density estimate (KDE) of the percentage of frames predicted to be a left turn and a right turn, respectively, by each annotator across all the videos. The major discrepancy between the two annotators is that A-2 systematically predicts a larger number of frames as

*Figure 5 continued on next page*

*Figure 5 continued*

behavior compared to A-1. (**E, H**) Confusion matrix showing the agreement between predictions of two classifiers over all the videos in the strain survey for left and right turn behavior. (**F, I**) Venn diagram capturing the frame-wise behavior agreement between the two annotators for left and right turn behavior.

image-recognition and action-detection fields, where an overlap of pixels of the bounding box and ground truth label box called the intersection over union (IoU; *Feichtenhofer et al., 2019*; *Kalogeiton et al., 2017*). We developed a graph-based approach called an *ethograph* to represent the bouts of behavior recorded in the ethograms of annotators A1 and A2. Concretely, we define the ethograph for two annotators as a bipartite graph $\mathcal{G} = (U, V, E)$, where $U$, $V$ are two disjoint sets corresponding to bouts predicted by each annotator and $E$ represents the edges that connect each element in set $U$ to an element in set $V$ capturing the overlap in time between the bouts. Further, the vertices of an ethograph represent bouts with vertex color encoding for the annotator and vertex size proportional to the duration of the bout. Further, the edges ($E$) of the graph ($\mathcal{G}$) represents the temporal overlap between the bouts (corresponding to different annotators) with the thickness of the edge proportional to the amount of bout overlap. *Figure 6B and C* shows the ethograms and their associated ethograph for the left turn behavior as predicted by annotators A1 and A2. In contrast to traditional frame-based ethograms, which simply display the sequential list of frames in which a behavior is observed, the ethograph allows for a more intuitive and visual representation of the temporal overlap between the bouts corresponding to different annotators (or even behaviors). This can be especially useful in identifying patterns and trends that may not be immediately apparent from comparing ethograms. By coloring the vertices and edges based on the annotator, it becomes easy to see which behaviors are consistently identified by both the annotators and which are more subjective and open to interpretation. Moreover, we can easily compute the bout-based agreement between the two annotators as the fraction of edges having thickness greater than some fixed threshold (see *Figure 6F* for mathematical definition) which essentially means the fraction of bouts having overlap greater than a chosen overlap threshold. The bout agreement between two annotators for the left and right turn at a threshold of 0.5 is shown as a Venn diagram in *Figure 6D and E* along with the density distribution of bout length. The agreement between two annotators with bout-based measure was certainly much better than that with frame-based comparison (see *Figure 5F and I*).

The predictions coming out of a classifier contained many short bouts (1–3 frames) of behavior that signal false positive bouts as they are much shorter than a typical bout of annotated behavior. Moreover, certain bouts of behavior were split by very short bouts (1–3 frames) of not-behavior signaling the presence of false negative bouts that result in fragmentation of a bout of behavior (see *Figure 6*). To address this issue, we proposed a stitching and filtering step on the predictions coming out of the classifier. First, we stitched those bouts whose distance to the neighboring bout is less than a certain fixed threshold. This stitched the fragmented bouts as illustrated in *Figure 6G*. We then applied bout filtering, which removed bouts of a length below a fixed threshold. To decide the optimal values of stitching and bout filtering thresholds, a hyper-parameter scan was performed for each behavior. *Figure 6H and I* present the results from hyper-parameter scan over stitching and bout filtering thresholds when the value of percentage bout overlap is fixed at 25%, 50%, and 75% for left (H) and right turn (I). *Figure 6J* captures the effect of applying bout filtering and stitching to a portion of an ethogram corresponding to the predictions made by A1 and A2 for the left turn behavior. The effect was clearly discernible when looking at the changes in ethograph, particularly with bouts (nodes) having multiple overlaps (edges) reducing to single overlap (edge) per bout.

In summary, when comparing classifiers, it is important to consider the inherent variability of human annotators. Frame-wise comparison penalizes this natural variability, making it a sub-optimal measure of agreement. On the other hand, bout-wise comparison takes this variability into account, making it a more biologically meaningful measure of agreement between classifiers. In addition to using bout-wise comparison, applying techniques like stitching and filtering can further improve agreement by reducing false and fragmented bouts in classifier predictions. By considering these factors, we can better understand the inter-annotator variability and design more effective guidelines for behavior annotation.

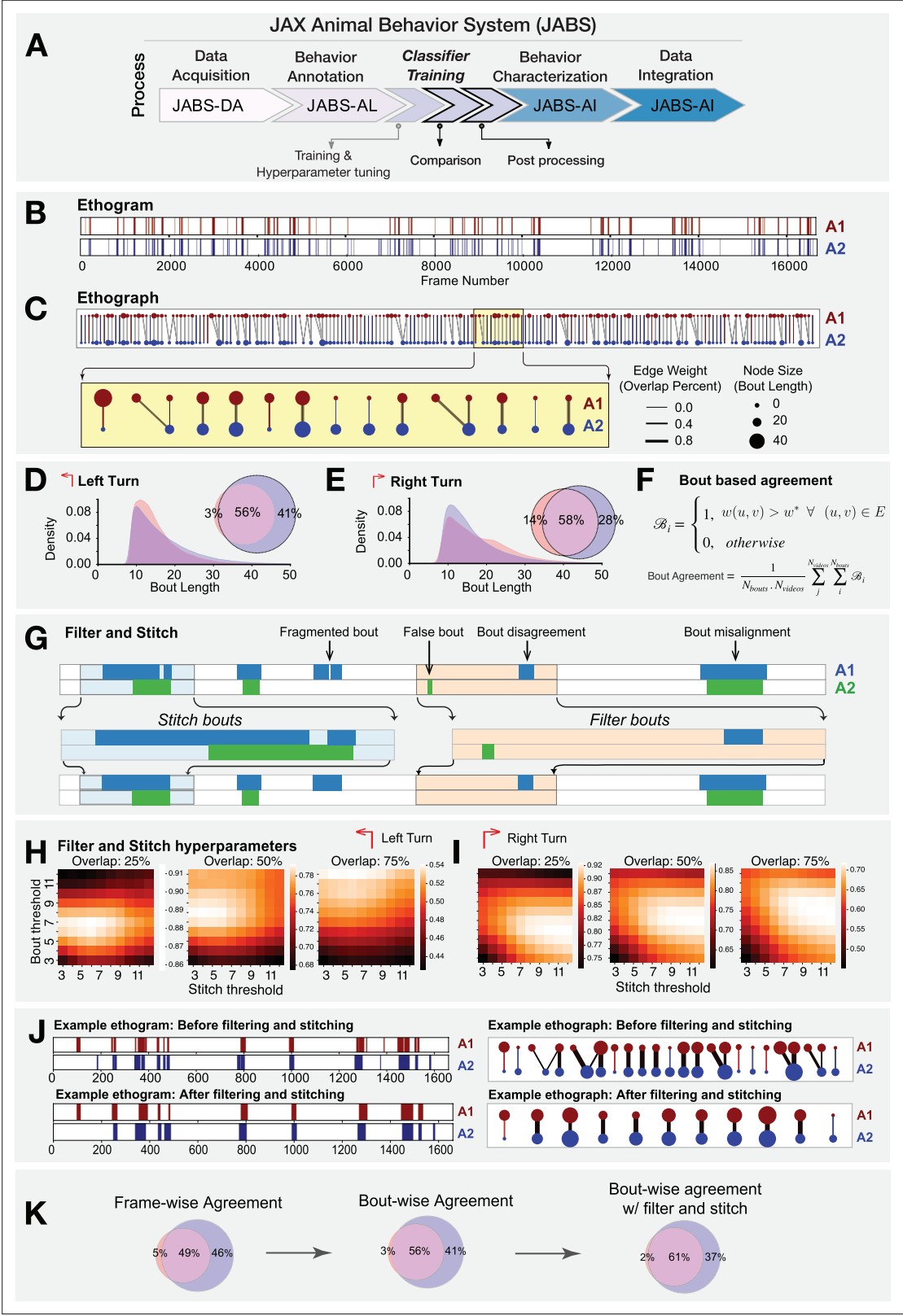

**Figure 6.** Bout-based comparison of classifier predictions from different annotators but trained for the same behavior. (**A**) JABS pipeline highlighting individual steps towards automated behavioral quantification. (**B**) Ethogram depicting frame-wise left turn predictions for annotators A1 (red) and A2 (blue). (**C**) Ethograph corresponding to the ethogram in panel (**B**) capturing the bout level information as a bipartite network. The nodes represent bouts with node size and color proportional to the bout length and annotator, respectively. Edge weights capture the fraction of bout overlap between

*Figure 6 continued on next page*

*Figure 6 continued*

two bouts predicted by different annotators for the same behavior. Edge weight and node size with zero value indicate missed bouts by an annotator. These have been given a small positive value for visualization purposes only. (**D–E**) Bout length distribution of annotators A1 and A2 for left and right turn behavior. (**F**) The mathematical definition of the average bout agreement between two annotators, where $w(u, v)$ represents weight between nodes $u$ and $v$ ($u \subset U$, $v \subset V$) in the ethograph $\mathcal{G}(U, V, E)$ and $w^*$ is the bout overlap threshold (0.5 fixed for our study). (**G**) Overview of the workflow for stitching and filtering at the bout level. (**H, I**) Hyper-parameters tuning to find optimal filtering and stitching thresholds. (**J**) Sample ethogram and its corresponding ethograph before and after applying stitching and filtering. (**K**) Inter-annotator agreement in frame-wise predictions underestimates the agreement, whereas the bout-wise comparison post filtering and stitching captures the overall agreement in a more biologically meaningful way.

## Compilation of strain survey datasets

In the present study, we have curated and are releasing three comprehensive datasets to the public, namely JABS600, JABS1200, and JABS-BxD, that encapsulate behavioral data derived from approximately 168 unique mouse strains, ensuring a balanced representation with a nearly equable distribution between female and male sex. The JABS600 dataset includes a total of 598 videos corresponding to 60 strains, approximately balanced with five males and five females per strain. On the other hand, the JABS1200 dataset contains 1139 videos corresponding to 60 strains, representing approximately nine males and nine females represented per strain. Finally, the JABS-BxD dataset includes a total of 1083 videos corresponding to 108 BxD strains that are derived from a cross between C57BL/6 J mice (B6) and DBA/2 J mice (D2). The duration of each video is approximately 1 hr, furnishing a substantial repository of behavioral data, which is invaluable for large-scale automated analysis of behavioral patterns. Furthermore, each video is supplemented with a corresponding keypoint file comprising 12 keypoints per frame, which is instrumental in extracting specific behavioral features. In line with our dedication to scientific openness and collaboration, we have made these datasets - encompassing both the video recordings and the keypoint files - available for public access at Harvard Dataverse (https://doi.org/10.7910/DVN/SAPNJG, https://doi.org/10.7910/DVN/RQYI04), making it easier for fellow researchers across labs to leverage our findings, replicate our experiments, and advance the field of automated behavior quantification.

## Strain survey of multiple behaviors

One of the advantages of a standardized data acquisition system such as JABS is that data can be repurposed. For instance, a classifier trained by another lab could be inferred on videos generated by another lab. We trained a set of behavior classifiers using the JABS active learning system and then inferred them on a previously published strain survey dataset (*Geuther et al., 2019*). The training dataset was composed of multiple human-annotated short videos (around 10 min each). We trained classifiers for left turn, right turn, grooming, rearing supported, rearing unsupported, scratch, and escape as examples. These can easily be extended to other behaviors. To capture the effect of genotype on the behavior, we subsampled the original strain survey data set to 600 one-hour open field videos representing 60 different strains with 5 female and 5 male for each strain and made predictions using the trained classifiers (see *Figure 7—figure supplement 2*). Further, we define three aggregate phenotypes associated with each behavior, namely the total duration of the behavior (in minutes) for the first 5, 20, and 55 min of the 1-hr video (*Geuther et al., 2021*), to capture the dynamic changes in behavior over time. The results are shown in *Figure 7B*, where the heatmap shows the Z-scores for the total duration of the behavior in 5, 20, and 55 min ($|Z\text{-score}| > 1$ thresholding is applied for easier visualization). The red and blue colored entries for a particular phenotype represent strains exhibiting the behavior that is more than one standard deviation above and below the mean of the phenotype respectively. Such data can have multiple utility. First, any user of JABS can conduct a rich analysis with little effort to yield biological insight. Such data can be used to refine classifiers by adding edge cases to training data. In addition, downstream genetic analysis such as heritability quantification and GWAS analysis are possible with this data (*Sheppard et al., 2022*; *Geuther et al., 2021*). In our analysis, we observed a high number of escape attempts in C58/J mice. This strain has been shown previously to have a high number of repetitive behaviors, perhaps even a strain for the study of autism features (*Ryan et al., 2010*; *Blick et al., 2015*; *Figure 7* Bottom panel). We find that other strains such as I/LnJ, C57/L, and MOLF/EiJ show increased levels of escape behaviors, thus increasing potential strains that could be used to model this behavior.

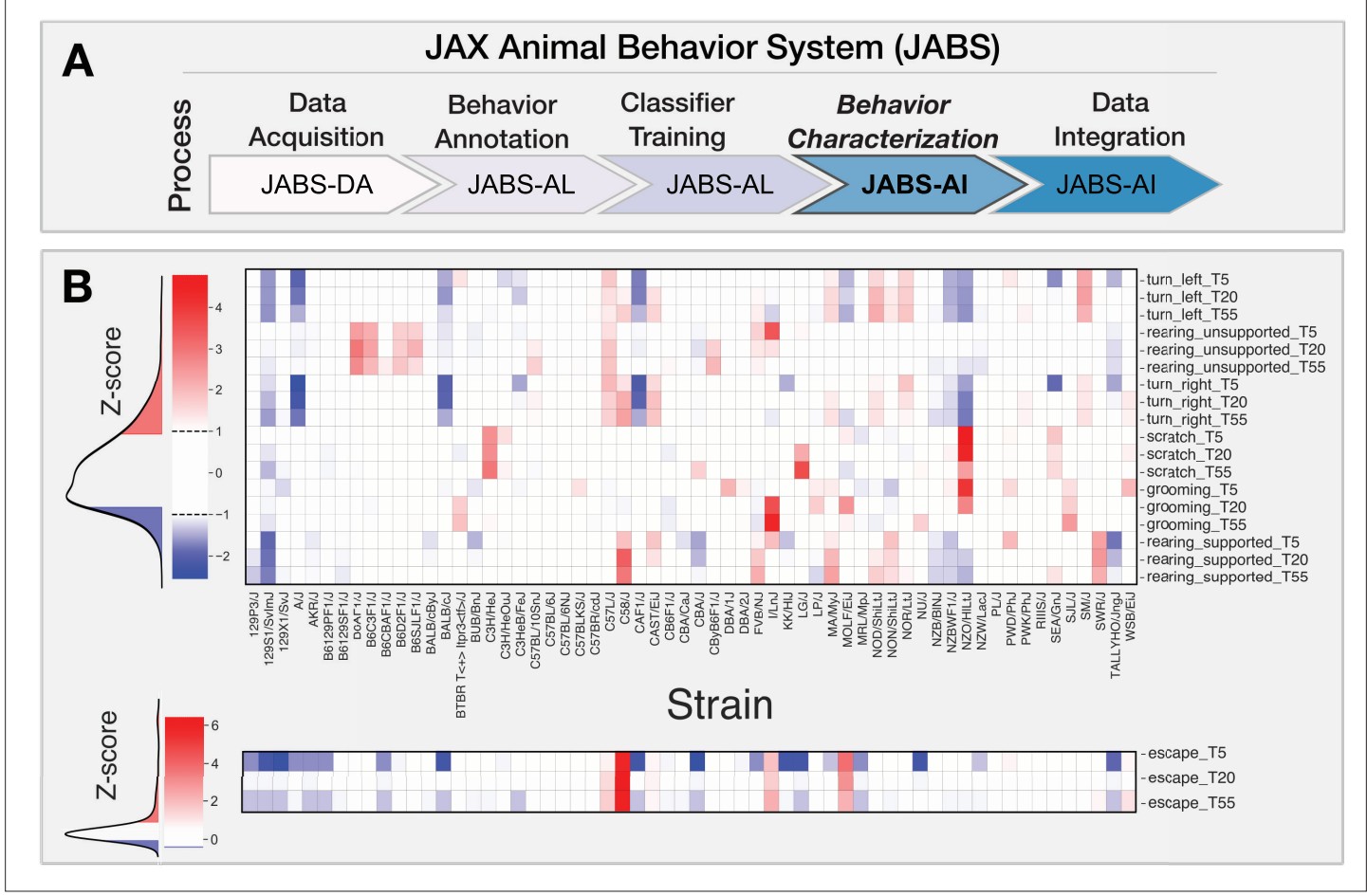

**Figure 7.** JABS-AI (analysis and integration) module: strain-level behavioral phenotyping across genetically diverse mouse populations. (**A**) JABS pipeline highlighting individual steps towards automated behavioral quantification. (**B**) Heatmap showing Z-transformed behavioral scores for aggregate phenotypes measured at three time points (5, 20, and 55 min) across the JABS600 strain survey. Each column represents a genetically distinct mouse strain, and each row corresponds to a specific behavioral measure including locomotion (turning left/right), exploration (rearing), self-directed behaviors (grooming, scratching), and escape responses. Color intensity indicates deviation from the population mean, with red representing increased behavioral expression and blue representing decreased expression relative to the strain average. Z-score thresholding (|Z-score|>1) was applied to all behavioral measures, with escape behaviors displayed separately using modified thresholding parameters to preserve detection of outlier strains exhibiting rare but phenotypically important escape responses. Behavioral measures are stratified by time point (**T5, T20, T55**) to capture temporal dynamics of phenotypic variation across genetic backgrounds.

The online version of this article includes the following figure supplement(s) for figure 7:

**Figure supplement 1.** JABS behavior characterization module: univariate analysis captures the combined effect of sex and strain on the aggregate phenotypes using the JABS600 dataset.

**Figure supplement 2.** JABS 600 strain distribution by sex.

In addition to phenotypic diversity due to genotype, we explored sexual dimorphism in our dataset with these new classifiers. We examined the impact of sex on the aggregated phenotype in various strains using a univariate approach. To test for the statistical significance of the effect of sex, we utilized a nonparametric rank test and corrected for multiple testing using false discovery rate (FDR) using Benjamini-Hochberg method. The LOD scores and effect sizes are presented in *Figure 7— figure supplement 1*, with the left panel showing the strength of evidence against the null hypothesis of the non-sex effect. The right panel presents a representation of the direction and magnitude of the effect size, with the color and size of the circle representing the direction and magnitude of the effect, respectively. The strains highlighted in pink exhibit a significant sex effect for at least one of the aggregated phenotypes. It is important to note that we are generally underpowered with five animals of each sex. However, we find that a high proportion of phenotypes show a sex effect.

## JABS-AI (analysis and integration) module: heritability, genetic correlation, and GWAS analysis

Next, one of our goals is to understand the genetic architecture that governs the complex behavioral phenotypes. The quantitative-genetic analysis described in this section provides the backbone for the JABS-AI web platform that follows in section Data Integration: A web application for classifier sharing and downstream genetic analysis. The web interface lets any user compute behavioral predictions and downstream genetic analysis for newly uploaded classifiers.

To facilitate this, we utilized the data derived from 49 inbred strains along with 11 F1 hybrid strains to perform a genome-wide association study (GWAS). It was deemed necessary to exclude the six wild-derived strains due to their pronounced divergence, which carried the risk of distorting the outcomes of our mouse GWAS. We first carried out power analysis for both the strain survey datasets (JABS600, JABS1200) using simulation algorithm as proposed by Genome-wide Efficient Mixed Model Association (GEMMA) software. GEMMA, a useful tool for this type of analysis, accounts for population structure and genetic relatedness between individuals, making it ideal for our inbred and hybrid strains. The power analysis as shown in *Figure 8A* revealed that we had sufficient statistical power to detect genetic associations. Notably, the JASB1200 demonstrated higher power in detecting these associations compared to the JABS600 dataset. With JABS1200 established as our dataset of choice for conducting the GWAS (see *Figure 8—figure supplement 1*), we moved forward with assessing each of the 72 phenotypes for their potential association with genotype. We employed the GEMMA software for this purpose, giving particular emphasis to the Wald test p-value in our analysis. These 72 phenotypes have been derived from eight basic classifiers, which include turn left and turn right (each assessed by two different annotators), grooming, scratching, supported rearing, and unsupported rearing. Each of these classifiers has been further categorized into three bout-based measures: average bout length, total duration, and total number of bouts. These bout-based measures were then dissected into three time-based measures (5 min, 20 min, and 55 min) to provide a comprehensive analysis. We tested a substantial number of SNPs (211,077) which necessitated accounting for the inherent correlations among SNP genotypes. To establish an empirical threshold for the p-values, we shuffled the values of one normally distributed phenotype (*TL_T*20_*duration*) randomly and identified the smallest p-value from each permutation. This rigorous process allowed us to set a p-value threshold of 1.9e-05 that reflects a corrected p-value of 0.05. We first report the heritability estimates for phenotypes corresponding to 55 min of observed behavior as shown in *Figure 8B*. Most of the phenotypes have heritability in the range (0.2–0.8) with bout length-based phenotypes having lower heritability relative to bout number or bout duration-based phenotypes. Next, to further shed light on the pleiotropic action of genes, we estimate the genetic correlations across these phenotypes using the bivariate linear mixed model implemented in GEMMA. We plot the genomic restricted maximum likelihood (GREML) estimates of bivariate genetic correlations in *Figure 8C*. The magnitude of the genetic correlation estimate provides an estimate of genetic overlap (common genetic loci) between two traits, whereas the sign determines the direction of the effects of the overlap on the two traits, that is a negative sign corresponds to the effect in the opposite direction on the two traits and vice versa. We hypothesize that for a given behavior, the bout-based measures within the behavior share common genetic effects and affect the traits in the same direction. Indeed, we find estimates of genetic correlations that are positive between the number of bouts (nBouts) and duration of the behavior (duration), the average length of each bout (avgLen) and duration of the behavior (duration), and the number of bouts (nBouts) and duration of the behavior (duration) for all behaviors except the turn right behavior (A1_TR) by annotator 1. We find positive estimates of genetic correlations between two annotators (A1, A2) for the turn left or right behaviors since we expect the genetic architecture underlying the same behavior from two annotators to overlap maximally.

We adopted a specific approach to identify quantitative trait loci (QTL): we started with the SNP that exhibited the lowest p-value across the genome and designated it as a locus. We then grouped together adjacent SNPs showing a significant level of correlation in their genotypes ($r^2 \geq 0.2$), employing a greedy strategy. We continued this process, moving on to the next SNP with the lowest p-value until we allocated all significant SNPs to a QTL. Given the inherent genetic structure of inbred mouse strains, large linkage disequilibrium (LD) blocks are expected, as represented in *Figure 8D*.

Additionally, we observe pleiotropy with certain loci displaying significant associations with multiple phenotypes, an anticipated occurrence given the correlation among many of our phenotypes and

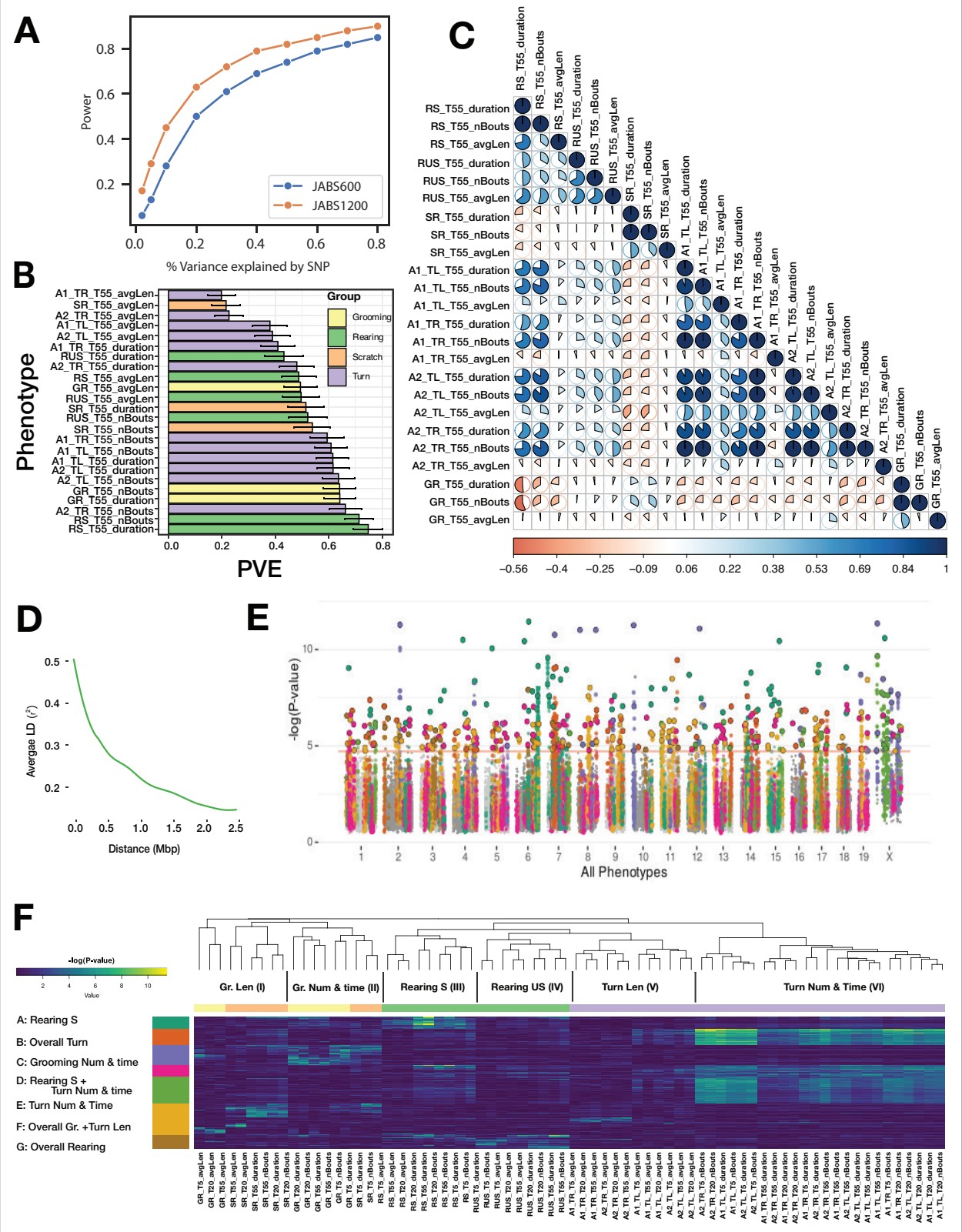

**Figure 8.** JABS-AI (analysis and integration) module: large-scale GWAS investigation of different mouse behaviors utilizing the JABS1200 dataset. (**A**) Statistical power comparison between two datasets (JABS600 vs JABS1200) at the genome-wide significance threshold of 2.4e-07. The y-axis shows how power varies with SNP effect size (x-axis). (**B**) Aggregate (55 min) phenotypes' heritability (PVE) estimates. (**C**) Lower Triangular Matrix Representation of Genotypic Correlation Among all of the 55-min aggregate phenotypes using a bi-variate linear mixed model, (**D**) Linkage

*Figure 8 continued on next page*

*Figure 8 continued*

disequilibrium (LD) blocks size, along with the mean genotype correlations for SNPs at varying genomic distances. (**E**) Aggregated GWAS results graphically represented via a comprehensive Manhattan plot. Peak SNP clusters, extracted from (**F**), determine color differentiation; SNPs within the same LD block are color-coordinated to match their peak SNP. Each SNP is assigned the minimum p-value derived from all phenotypes. (**F**) An inclusive heatmap exhibiting all the significant peak SNPs for each phenotype. Each row, representing an SNP, is color-coordinated according to the allocated cluster within the k-means clustering. The color scheme originating from the k-means cluster is also applied in panel E of this analysis.

The online version of this article includes the following figure supplement(s) for figure 8:

**Figure supplement 1.** JABS 1200 strain distribution by sex.

the potential for individual traits to be influenced by similar genetic loci. To get a clearer picture of the pleiotropic structure apparent in our GWAS findings, we constructed a heatmap (***Figure 8F***) of significant QTL across all phenotypes and employed *kmeans* clustering to identify QTL sets governing phenotype groups. The phenotypes are grouped into six categories, namely: grooming bout length, grooming bout number and total time, rearing supported, rearing unsupported, turn bout length, turn bout number, and total time. We uncovered seven unique clusters of QTLs (A-G), each regulating a different combination of these phenotype subgroups (***Figure 8F***). Clusters B and G notably held pleio-tropic QTLs that influenced overall turn and rearing behaviors, respectively. Yet within cluster F, we identified distinct QTL sets - one that steered grooming behavior, and another, non-overlapping set that determined turn length. This distinction signifies the existence of distinct genetic underpinnings for these different behaviors even within the same cluster. Finally, we color the associated SNPs in the Manhattan plot (***Figure 8E***) showing QTLs associated with all phenotypes.

## Data integration: a web application for classifier sharing and downstream genetic analysis

In conjunction with the release of the curated datasets and the JABS active learning GUI app (JABS-AL), we have developed and launched a web-based application, JABS-AI (analysis and integration), aimed at streamlining the sharing and utilization of classifiers. Through this platform, users can view, download, and rate the classifiers for various behaviors that have been developed and trained in our laboratory as shown in ***Figure 9***. In addition, it provides an insight into their heritability scores and offers a feature to examine the pair-wise genetic correlations amongst different phenotypes. An added functionality of this web application is that it allows users to upload their own classifiers (trained using JABS-AL GUI app) for any specific behavior. Upon uploading, the application automatically executes the classifier on a dataset of the user's choosing from our strain survey datasets. It conducts an automated analysis of behavior and genetics and subsequently dispatches the results to the user's designated email address within a few hours (see ***Figure 9A***). The results coming out of the web app would contain the following downstream analysis on the dataset selected by the user in the app:

1. Density plots of predicted behavior (similar to ***Figure 5D and G***).
2. Strain-behavioral phenotype heatmap (similar to ***Figure 7***).
3. Heritability and genetic correlation of behavioral phenotypes (similar to ***Figure 8B and C***).

This web application serves as a facilitative tool aimed at fostering collaboration among researchers and streamlining the advancement of automated behavior quantification studies by providing a plat-form for the efficient sharing and analysis of behavioral classifiers.

## Discussion

Democratization of machine vision methods for advanced behavior quantification remains a challenge. Often, tracking and behavior classifiers are not transferable between laboratories. This limits the reuse of prior work with each laboratory essentially starting from scratch with advanced behavior quantifi-cation. JABS and the companion DIV Sys are designed to overcome these limitations. JABS compo-nents include video data acquisition, behavior annotation, classifier sharing, and genetic analysis. By adopting the JABS-DA laboratories, can use our pose estimation and segmentation models that work across 62 mouse strains of varying coat colors and sizes. This greatly eases the barrier to entry for advanced behavior quantification. The next steps of creating behavior classifiers are carried out using JABS-AL, an active learning system modeled after JAABA (***Kabra et al., 2013***). We benchmarked

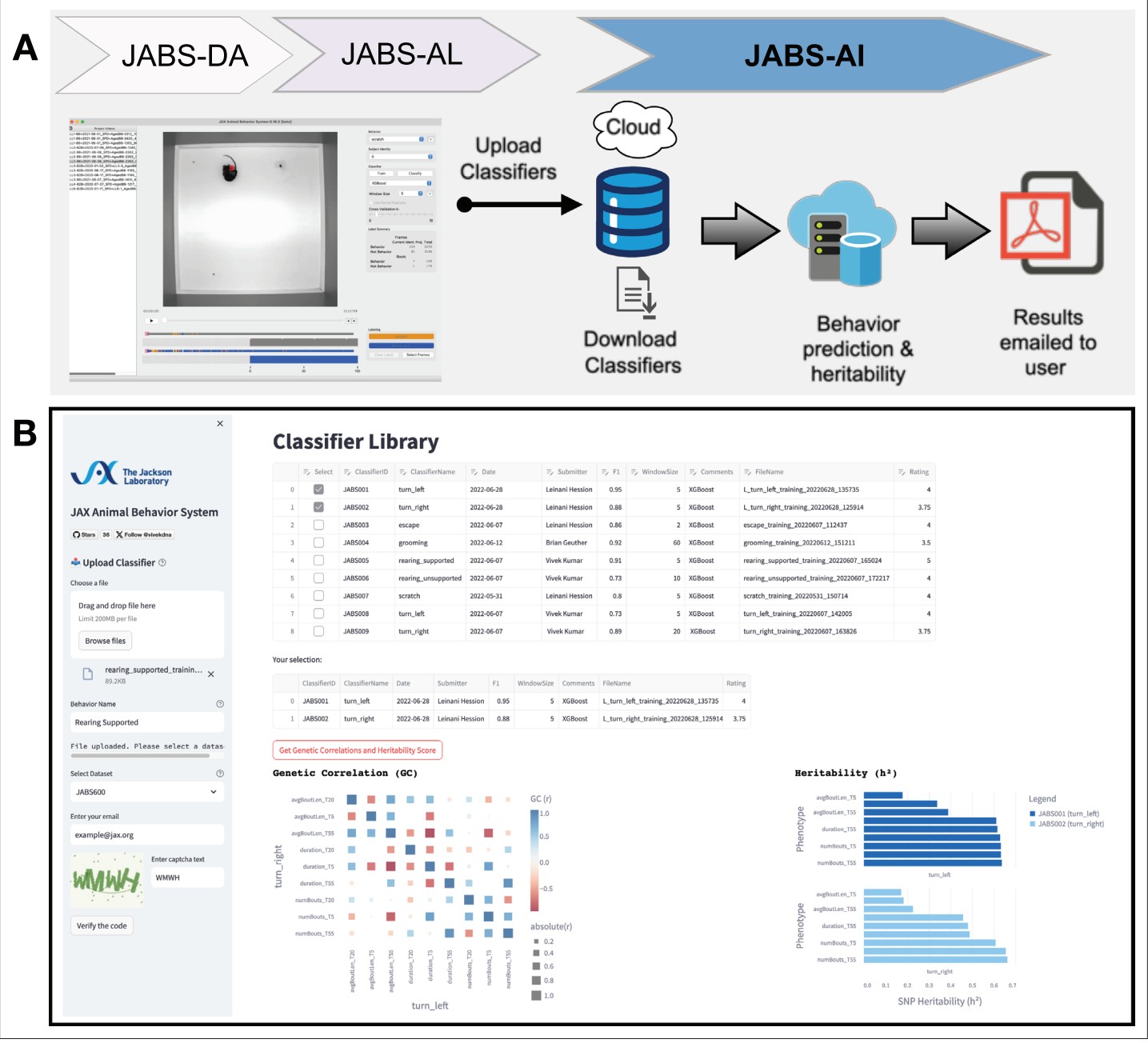

**Figure 9.** JABS-AI (analysis and integration): a web application for sharing the JABS classifiers and automated downstream genetic analysis. (**A**) Illustrates the fundamental workflow of the web application, beginning with the user employing a classifier trained via the JABS active learning application. The user subsequently deposits this classifier into our web application, which performs comprehensive automated analyses, encompassing both behavioral and genetic aspects, on the dataset selected by the user from our curated strain survey collection (JABS600, JABS1200) accessible via a dropdown menu. The outcome of these analyses, encapsulating detailed behavioral patterns and genetic correlations, is then dispatched to the user's designated email address within a short timeframe. (**B**) Screenshot of the web app highlighting the tabular presentation of the repository of classifiers developed in our laboratory, complete with pertinent metadata such as the date of creation, training hyperparameters, and user ratings. When any two classifiers are selected, the application offers the option to analyze the genetic correlations between the phenotypes corresponding to the selected classifiers, in conjunction with their heritability scores.

JABS-AL using the grooming data set and demonstrated that it reaches very good performance with 10% of the data needed for a 3D-CNN for action detection. Once constructed, behavior classifiers can be shared through JABS-AI, a cloud-based tool. Labs can create their own behavior classifiers using JABS-AL or download an existing one for use from another lab from JABS-AI in order to annotate

single behaviors. The power of JABS-AI is also in the embedded strain survey data. A deposited classifier is inferred on one of three datasets, and heritability and genetic correlation results are returned.

A key decision point is the adoption of common apparatus to create a uniform visual look of the video data across laboratories. This enables cross-application of foundational models and exchange of behavior classifiers across labs. We realize that this may be challenging for some labs with limited space and budget. Indeed, JABS has a large footprint with a 2x2 × 6 feet (W x L x H) space requirement, costs several thousand dollars for components, and requires some computational expertise to set up and operate. Laboratories must balance this cost with the labor and time costs of adopting an existing set-up for advanced behavior analysis. In lieu of adopting a common apparatus, efforts are being made to build foundational models that can handle diverse environments and even animals (*Ye et al., 2024*). While similar datasets are common in human pose estimation, the development of equivalent datasets for animals is still underway. However, such foundational models are not yet available. Even when they are available, the initial abstraction step is simple compared to the later step of behavior classification. For instance, in our gait and posture paper, producing a pose estimation model that generalizes to diverse mouse strains took approximately 6 months. It was an iterative hard example mining task. However, the process of deriving gait and posture from keypoints and its genetic validation took over 1.5 years. By simply adopting JABS, laboratories gain access to both the pose model and validated gait/posture algorithms.

While our pose estimation model was not specifically trained on tethered animals, published research demonstrates that keypoint detection models maintain robust performance despite the presence of headstages and recording equipment. Once accurate pose coordinates are extracted, the downstream behavior classification pipeline operates independently of the pose estimation method and would remain fully functional. We recommend users validate pose estimation accuracy in their specific experimental setup, as the behavior classification component itself is agnostic to the source of pose coordinates.

Another added benefit to JABS is the ability to apply novel behavior classifiers to large-scale genetically diverse datasets collected at JAX through JABS-AI. We have modeled this after existing platforms such as GeneNetwork and DO QTL. Currently, we provide heritability estimates of any classifier that is deposited. Users can also select behaviors to genetic correlation studies. Thus, even if two behaviors are different, they may measure the same genetic architecture. Although the current version of JABS-AI does not offer GWAS analysis due to compute restrictions, the method can be easily extended for such analysis. It is also feasible to link animal behaviors to human traits through PheWAS analysis (*Pendergrass et al., 2011*; *Geuther et al., 2019*). This would provide even more detailed information for users about the genetic regulators of complex behaviors. The current data sets, JABS600, JABS1200, and JABS-BxD consist of young wild-type animals. We have collected datasets in aging populations with various frailty statuses and animals that display nocifensive behaviors. These could also be integrated into JABS-AI for preclinical behaviors independent of genetic analysis.

While JABS is designed for individual behavior annotation, a common task in behavioral neurogenetics to determine an internal state, e.g. anxiety or social state. Often, these are accomplished by measuring a single behavior. A more powerful approach is the application of behavior indices to predict certain states. These indices can be constructed using multiple behaviors and even other covariates. For example, we trained a model to predict frailty using data from over 600 JABS-DA open field tests from C57BL/6J mice of varying age and frailty. We used 34 features from JABS to derive frailty. Similarly, in companion work, we derived a pain scale with 82 features from JABS. These indices were constructed with almost 1000 animals and can readily be transferred to other labs that collect data using the JABS system. This is incredibly powerful and allows labs to leverage each other's models by using a common platform. Similarly, for pain states, we have tested multiple strains and built pain intensity models that can be utilized. We believe a true advantage of advanced phenotyping using video data is the ability to reuse and extract more information from existing data. This essentially allows us to use fewer animals, a core 3R principle.

Here, we describe JABS as a single animal open-field assay that lasts from minutes to a few hours. However, we have designed the JABS arena for long-term housing of animals with a food hopper and lixit. This was the primary reason we worked with JAX-IACUC to certify JABS-DA for long-term monitoring with key required environment measures. By blocking visible light and imaging using IR LED illumination, we obtain uniform data at night or day. We routinely collect video data with three mice

over several days. The models for tracking, instancing, and identity maintenance need to evolve. We plan to extend JAB-AI with classifiers for social interactions and homeostatic behaviors. Thus, future iterations of JABS will develop and share multi-animal behavior analysis.

Even when data acquisition is standardized, another fundamental source of variability can enter the system when different human experts within or across different labs annotate the same videos for the same behavior. This type of variability can arise due to a variety of factors, including differences in training, personal biases, and individual interpretation of behavior. As behaviors become more complex, we expect behaviorists to show more disagreement. These disagreements could be as simple as varying understanding of the starts and stops of the behaviors. Or more fundamentally, differing opinions on the behavior such as between aggression and play. In a previous study, we asked five humans to annotate grooming behavior and found that the agreement ranged from 86% to 91%. In this case, we simply compared on a frame-wise basis labels from annotators who were asked to label every frame in the same set of videos. In most cases, such comparisons are infeasible. A more realistic comparison is provided in this manuscript. Two annotators built their own classifiers for left and right turn behaviors. Then we compared the predictions from each set of classifiers on JABS-DA video. This is akin to two different laboratories that may deposit classifiers for the same behavior. JABS-AI does not save primarily the training data from each annotator (lab) and simply uses the trained classifiers to infer on a new set of videos (e.g. JABS600). From these, we can compare the overlap between the two classifiers' inferences. We do not make assumptions about which classifier is the ground truth and simply compare both classifiers.

In the section Frame and bout-wise classifier comparison of inter-annotator variability, we demonstrate that even for simple behaviors like left and right turn, there is a significant amount of disagreement between predictions coming from classifiers trained by two expert annotators within the lab for the same behavior. One of the most commonly used statistical measures to quantify the inter-annotator variability is Cohen's kappa, which assesses the level of agreement between the annotators, taking into account the possibility of agreement by chance. The Cohen's kappa statistic works well for frame-wise comparison but is ill-defined for bout-wise comparison, as unlike frames, bouts are not conserved. In order to overcome this limitation, we have introduced a new approach based on graph theory, called the *ethograph*. This network approach allows us to define measures that quantify the agreement between two annotators when comparing bouts of behavior among different annotators. By comparing the entire sequences of frames, the ethograph reduces subjectivity and allows for a more holistic and consistent interpretation of behaviors. This makes it well-suited for bout-wise comparison and may provide a more accurate estimate of inter-annotator agreement than the frame-based kappa statistic.

Even though frame-wise comparison shows the overlap performance is poor ($\kappa$=0.64 and 0.65 for Left and Right Turn, respectively), each classifier does a good job of identifying tuning behaviors. The turn behaviors are short in length of frames, and the two annotators differ in defining the starts and stops of the behaviors. One annotator labels just the core turn behavior, and the other starts labeling turning behavior a few frames earlier and ends later. Classifiers from both annotators generally find the same bouts of turning which we could visualize in the ethograph. We explored bout-wise accuracy metrics as an alternative to frame-wise metrics. We also explore post-processing predictions using hyperparameters for filter and stitch. By adding these, we observe much higher agreement between the classifiers from two annotators for the same behaviors (overlap increases from 49% to 61%). It is important that users clearly define the behavior as best as possible, and document the filter and stitch parameters.

JABS users, when confronted with multiple classifiers for the same behavior in JABS-AI, must prioritize use of one classifier. JABS-AI offers a genetic solution to this challenge - by prioritizing the classifier that is more heritable. Heritability is an estimate of variance explained by genetics and can act as a discriminator in this situation. We also calculate genetic correlation, which allows users to determine which underlying genetic construct is being measured. For instance, both left and right turns are highly genetically correlated and, therefore, for the purposes of genetics, there is simply turn behavior. However, for certain unilateral models such as brain lesions, stroke, optogenetic stimulation, or injury, the ability to distinguish left and right turns can be critical.

## Supervised vs unsupervised behavior segmentation

Unsupervised methods like Keypoint-MoSeq (*Weinreb et al., 2024*), VAME (*Luxem et al., 2022*), B-SOiD (*Hsu and Yttri, 2021*), and MotionMapper (*Berman et al., 2014*) which prioritize motif

discovery from unlabeled data but may yield less precise alignments with expert annotations, as evidenced by lower F1 scores in comparative evaluations (*Blau et al., 2024*). Supervised approaches (like ours), by contrast, employ fully supervised classifiers to deliver frame-accurate, behavior-specific scores that align directly with experimental hypotheses. Ultimately, a pragmatic hybrid strategy, starting with unsupervised pilots to identify motifs and transitioning to supervised fine-tuning with minimal labels, can minimize annotation burdens and enhance both discovery and precision in ethological studies.

## Future directions and challenges

We see several areas of improvement in JABS in the future. First, the success of such a platform depends on community adoption. As such, JAX has made JABS free for noncommercial use, and we have listed all parts and software used to make JABS. We realize that many laboratories may not have the computational or fabrication resources to construct JABS and that commercial suppliers who can provide a turnkey system are needed for JABS-DA. JABS-AL and JABS-AI require fewer, although still significant, resources to support.

JABS-AI currently does not support the upload of training videos due to resource limitations. This prevents other users from interrogating the primary training labels. It also prevents users from downloading and labeling new behaviors or modifying classifiers that have been uploaded. Future versions could support sharing of complete training data instead of the classifier only.

Furthermore, since the classifiers are trained on few densely labeled short video recordings and then further make predictions on a large strain survey consisting of multiple strains of mice, there is some variability in predictions purely due to out-of-distribution strains in the strain survey. Therefore, the inter-annotator variability in predictions on the new set of strains of mice can be attributed to both the variability in the human labeling and genetic variability in the strain survey. Calculating the heritability scores might help in this scenario by providing us a quantitative measure of the extent to which the inter-annotator variability is due to genetic factors versus interpretation by the human labelers.

### Rodent homes and hotels

Finally, JABS and DIV Sys are complementary systems that enable behavioral monitoring across multiple scales and resolutions. DIV Sys facilitates long-term observation of home-cage behaviors, whereas JABS offers high-resolution tracking of gait, posture, and other discrete actions (*Sheppard et al., 2022*; *Geuther et al., 2021*). The larger space in JABS can potentially accommodate additional tasks designed to probe specific neural circuits (*Rosenberg et al., 2021*; *Arakawa et al., 2007*), and neural recordings can be collected from instrumented mice in this same arena. We see these as 'ethological tasks' that can be performed continuously over long periods of time in order to interrogate neural and genetic circuits in customizable environments – 'hotels'. Examples include mazes and other tasks that neurobehavior researchers have been developing. These assays can be validated using genetic or pharmacological models on a shared platform such as JABS. These two platforms provide a dual approach: continuous surveillance of mice in their home-cage environments (via DIV Sys) alongside targeted assessments of particular behaviors in a dedicated 'hotel' arena (via JABS). This combined paradigm presents a powerful framework to link genetic and neural changes to complex behaviors. Indeed, elucidating how altered behaviors result from altered neural circuits and altered genetic pathways remains a central challenge in computational ethology — one that platforms such as JABS and DIV Sys are poised to address.

## Materials and methods
### Animals

All animals were obtained from The Jackson Laboratory production colonies or bred in a room adjacent to the testing room as previously described (*Geuther et al., 2019*; *Kumar et al., 2011*; *Geuther et al., 2021*). All animal work was approved by The Jackson Laboratory Institutional Animal Care and Use Committee (AUS 14010). All JABS-DA open field arena work described here was approved under routine procedures VK14-01 and VK16-02. Mice received Lab Diet 5K52 6% sterilized grain and acidified drinking water ad libitum during standard housing before and after testing and during long-term JABS-DA (VK16-02). Mice in shorter 1 hr video observations, such as those in the data sets described

here, were without food, water, or bedding during the observation period (VK14-01). The complete table of all mouse strains used in this study and their JAX identifiers is listed in *Supplementary file 4*.

## Datasets

To facilitate reproducibility and community-driven discovery, we have made three comprehensive datasets publicly available. Each dataset contains multiple open-field arena (OFA) videos (one hour recording per video) and corresponding pose-estimation keypoint files.

### JABS600

This dataset comprises approximately 600 videos from 62 genetically diverse mouse strains, with sexes balanced within each strain. It serves as a broad survey for initial explorations of behavioral phenotypes across a wide genetic landscape https://doi.org/10.7910/DVN/SAPNJG.

### JABS1200

An extension of the first dataset, JABS1200 contains nearly 1200 videos from the same 62 strains, effectively doubling the sample size per strain. This increased depth provides greater statistical power for detecting significant associations in Genome-Wide Association Studies (GWAS) https://doi.org/10.7910/DVN/SAPNJG.

### JABS-BxD

This collection includes over 1000 videos from 108 BxD recombinant inbred mouse strains, which are derived from a cross between the C57BL/6 J (B6) and DBA/2 J (D2) parental strains. With approximately five males and five females per strain, this dataset is structured to support high-resolution genetic mapping of behavioral traits https://doi.org/10.7910/DVN/RQYI04.

### JABS behavioral classifier example projects

Complete training resources for behavioral classification, comprising experimental video data, extracted pose coordinates, and corresponding behavioral annotations, are made publicly available https://doi.org/10.5281/zenodo.16697332.

### Quantifying strain survey dataset imbalance

We present some metrics that enable us to capture the amount of the strain and gender imbalance in our datasets:

Strain Imbalance (SI):

$$SI = \max_{i=1}^{n} \left| \frac{n_i^m + n_i^f}{\sum_{j=1}^{n}(n_j^m + n_j^f)} - \frac{1}{n} \right| \tag{1}$$

Gender Imbalance (GI) for each strain $i$:

$$GI_i = \frac{|n_i^m - n_i^f|}{n_i^m + n_i^f} \tag{2}$$

The Average Gender Imbalance (AGI) can be calculated as the mean of the Gender Imbalance (GI) for all strains:

$$AGI = \frac{1}{n} \sum_{i=1}^{n} GI_i \tag{3}$$

1. $n$ is the number of strains
2. $n_i^m$ is the number of male samples for strain $i$
3. $n_i^f$ is the number of female samples for strain i.

## JABS workflow guide

The JAX Animal Behavior System (JABS) provides an integrated, four-stage pipeline guiding users from standardized data acquisition to novel genetic discovery. This workflow utilizes three core software modules (JABS-DA, JABS-AL, JABS-AI) and a standalone pose estimation engine, as detailed in the summary below.

| JABS end-to-end behavioral phenotyping workflow | | |
| --- | --- | --- |
| Step | Module and objective | Resources |
| Data acquisition | JABS-DA: capture uniform, high-fidelity video recordings to ensure cross-laboratory reproducibility. | Data acquisition pipeline |
| Pose estimation | Mouse tracking runtime: convert video into a quantitative representation of posture by tracking 12 body keypoints. | Tracking Runtime pipeline |
| Behavioral annotation and classifier training | JABS-AL: train supervised classifiers for user-defined behaviors using a sparse-labeling approach. | JABS-AL GUI app JABS-AL Tutorial |
| Genetic analysis | JABS-AI: deploy classifiers on large-scale genetic datasets to discover genetic drivers of behavior. | JABS-AI Webapp |

## Pose tracking pipeline

We have previously published our single mouse pose model here (*Sheppard et al., 2022*), with the training data and trained models available at https://zenodo.org/records/6380163. We have also released our pose tracking pipeline, which includes single and multimouse tracking and classifier prediction at https://github.com/KumarLabJax/mouse-tracking-runtime, *Geuther et al., 2026b*. In addition to our in-house tracking pipeline, we have made our pose format accessible via SLEAP-io conversion hooks (https://github.com/talmolab/sleap-io, *Pereira et al., 2026*).

## Quality control

In our released datasets, we provide video-level quality summaries for coverage of our pose estimation models. We currently maintain in-house post-processing scripts that handle quality control according to our specific use cases. Future releases of JABS will incorporate generalized versions of these scripts, integrating comprehensive QC capabilities directly into the platform. This will provide users with automated feedback on video quality, pose estimation accuracy, and classifier performance, along with diagnostic visualizations such as movement heatmaps and behavioral summary statistics.

## Grooming benchmarking study

The Convolutional neural network (CNN) applied to the grooming benchmark dataset follows a typical feature encoder structure except using 3D convolution and pooling layers instead of 2D. The final layer was used as the output probabilities for not grooming and grooming predictions for each frame. The exact architecture and the training details are described in detail here (*Geuther et al., 2021*). Furthermore, the JABS grooming classifier has been trained with both XGBoost (*Chen and Guestrin, 2016*) and Random Forest model (*Ho, 1995*) with their default hyper-parameters in the XGBoost and scikit-learn library, respectively. To assess classifier performance on frame-by-frame labeling of animal behavior, we report five standard metrics:

### Accuracy

Percentage of all video frames correctly labeled as either target behavior or not.

$$\text{Accuracy} = \frac{TP + TN}{TP + TN + FP + FN}$$

## F1 score

Harmonic mean of precision (fraction of predicted target frames that are correct) and recall (fraction of actual target frames detected).

$$\text{F1} = 2 \times \frac{\text{Precision} \times \text{Recall}}{\text{Precision} + \text{Recall}}, \qquad \text{Precision} = \frac{TP}{TP+FP}, \quad \text{Recall} = \frac{TP}{TP+FN}$$

## AUROC

Area under the receiver-operating-characteristic curve; measures how well the classifier separates frames containing the target behavior from all others, independent of the decision threshold (k).

$$\text{AUROC} = \int_0^1 \text{TPR}(k) \, d(\text{FPR}(k))$$

## True positive rate (TPR)

Proportion of frames with the target behavior that the classifier correctly labels.

$$\text{TPR} = \frac{TP}{TP+FN}$$

## False positive rate (FPR)

Proportion of frames without the target behavior that are incorrectly labeled as target by the classifier.

$$\text{FPR} = \frac{FP}{FP+TN}$$

## TPR@5 % FPR

In practical applications, it is important to capture as many true instances of the behavior as possible while avoiding excessive false positives, which reduce trust in the predictions and increase validation effort. Therefore, we also report the TPR achieved when the classifier is tuned to allow at most 5% FPR.

TP = true positives, TN = true negatives, FP = false positives, FN = false negatives; all defined at the frame level.

## Downstream analysis on strain survey

### Aggregate phenotypes

For each classifier, we construct nine aggregate phenotypes corresponding to the three metrics (total duration, number of bouts, and average bout length) and three time bins (first 5, 20, and 55 min). For instance, a phenotype named 'turn_left_T55' (in *Figure 7*) represents the total duration of left turn behavior in the first 55 min of the video averaged across all videos. For more details, refer to *Supplementary file 2*.

### Z-score normalization of behavioral phenotypes

To facilitate comparison across strains and behaviors, each phenotype was standardized using z-score normalization. This process transforms the raw phenotype values into units of standard deviation relative to the mean across all strains. Specifically, for each phenotype $x$, the z-score $z$ is calculated as

$$z = \frac{x - \mu}{\sigma}$$

where μ is the mean and σ is the standard deviation of the phenotype values across all strains. This normalization centers the data around zero and scales it so that the spread reflects variability within the dataset. In our heatmap (*Figure 7*), colors correspond to z-scores, emphasizing how much a strain's behavior deviates above (red) or below (blue) average. To improve clarity, we highlight entries with $|z| > 1$, indicating behaviors differing from the mean by more than one standard deviation.

## Genetic analysis

The aggregate behavioral phenotypes were analyzed to study the genetic associations of strain-specific behaviors. Genotype data for the different mouse strains were obtained from the Mouse Phenome Database (https://phenome.jax.org/genotypes). We used genotypes derived from the Mouse Diversity Array (MDA), where di-allelic genotypes were inferred from parental genomes. Quality control filters retained SNPs with a minor allele frequency (MAF) of at least 10% and a maximum of 5% missing data.

Genome-wide association studies (GWAS) were conducted using the R package mousegwas, as previously described in *Geuther et al., 2021*. Classical laboratory mouse strains were included in the analysis, excluding wild-derived strains. Associations were computed using the linear mixed model (LMM) method implemented in GEMMA. To reduce confounding from linkage disequilibrium near tested markers, a Leave One Chromosome Out (LOCO) approach was applied: for each chromosome under test, the kinship matrix was calculated using SNPs from all other chromosomes.

To control for multiple testing and establish an appropriate genome-wide significance threshold, we performed permutation-based empirical calibration. Specifically, phenotype values from a randomly sampled continuous trait were shuffled across individuals multiple times, and the minimum p-value from each permutation was recorded. This approach yielded a p-value threshold of approximately $1.9e{-}05$, corresponding to a family-wise error rate of 0.05 after correcting for the number of tests performed.

SNP-based heritability for each behavioral phenotype was also estimated using mousegwas. For each phenotype, a genetic relatedness matrix (GRM) was constructed from quality-controlled SNPs (filtered for MAF and missingness). The analysis was performed on the JABS1200 dataset, comprising 1139 individuals and 211,070 SNPs. Fixed-effect covariates included sex, weight, and coat color of the animals.

The GWAS execution was wrapped in an R package called mouseGWAS available on GitHub: https://github.com/TheJacksonLaboratory/mousegwas, *Peer and Kohar, 2021*.

For genetic correlation analyses, we applied GEMMA's bivariate LMM (using this pipeline: https://github.com/gautam-sabnis/genetic_correlation) *Sabnis and anshu957, 2023* to each pair of phenotypes.

## Power analysis simulation for GWAS

To assess the statistical power of our GWAS design, we followed the simulation-based approach described in the original GEMMA paper (*Zhou and Stephens, 2012*). Briefly, we first filtered SNPs from the GEMMA output (for a randomly selected behavioral phenotype) to retain only those with nominal association ($p < 0.05$), sorted them by genomic position, and then selected a fixed number of evenly spaced SNPs as 'causal'. For each causal SNP, we assigned an effect size required to explain a specified proportion of phenotypic variance (PVE), calculated as $\text{effect size} = \sqrt{\frac{\text{PVE} \times \text{Var(phenotype)}}{\text{AF} \times (1 - \text{AF})}}$, where AF is the SNP allele frequency. We simulated new phenotypes by adding the effect of each causal SNP to the original phenotype, re-ran GWAS using GEMMA, and defined power as the proportion of simulated causal SNPs detected above the genome-wide significance threshold (e.g. Bonferroni-corrected $p < 0.05$).

# Acknowledgements

We thank members of the Kumar Lab for helpful advice and Leinani Hession for training behavior classifiers. Michelle Foskett (Process Quality Control) and Rosalinda Doty (Diagnostic and Pathology Services) help with environment and pathology data. This work was funded by The Jackson Laboratory Directors Innovation Fund, National Institute of Health DA041668 (NIDA), DA048634 (NIDA), MH138309 (NIMH), and AG078530 (NIA). All code and training data will be available at Kumarlab.org and Kumar Lab Github (https://github.com/KumarLabJax).

## Additional information

### Funding

| Funder | Grant reference number | Author |
|---|---|---|
| National Institutes of Health | DA041668 | Vivek Kumar |
| Jackson Laboratory | Director's Innovation Fund | Vivek Kumar |
| National Institutes of Health | DA048634 | Vivek Kumar |
| National Institutes of Health | MH138309 | Vivek Kumar |
| National Institutes of Health | AG078530 | Vivek Kumar |

The funders had no role in study design, data collection and interpretation, or the decision to submit the work for publication.

### Author contributions

Anshul Choudhary, Conceptualization, Resources, Data curation, Software, Formal analysis, Validation, Investigation, Visualization, Methodology, Writing – original draft, Writing – review and editing; Brian Q Geuther, Resources, Software, Formal analysis, Investigation, Methodology, Writing – original draft; Thomas J Sproule, Resources, Investigation, Methodology, Writing – original draft; Glen Beane, Conceptualization, Resources, Data curation, Software, Investigation, Visualization, Writing – original draft; Vivek Kohar, Data curation, Formal analysis, Investigation; Jarek Trapszo, Resources, Investigation; Vivek Kumar, Conceptualization, Supervision, Funding acquisition, Visualization, Writing – original draft, Project administration, Writing – review and editing, Methodology

### Author ORCIDs

Anshul Choudhary ⓘ https://orcid.org/0000-0001-6651-5224
Brian Q Geuther ⓘ https://orcid.org/0000-0002-7822-486X
Vivek Kohar ⓘ https://orcid.org/0000-0003-1813-1597
Vivek Kumar ⓘ https://orcid.org/0000-0001-6643-7465

### Ethics

All animal work was approved by The Jackson Laboratory Institutional Animal Care and Use Committee (AUS 14010). All JABS-DA open field arena work described here was approved under routine procedures VK14-01 and VK16-02.

Reviewer #1 (Public review): https://doi.org/10.7554/eLife.107259.3.sa1
Reviewer #2 (Public review): https://doi.org/10.7554/eLife.107259.3.sa2
Author response https://doi.org/10.7554/eLife.107259.3.sa3

## Additional files

### Supplementary files

MDAR checklist

Supplementary file 1. Training and classifier metadata for grooming benchmark. Table 1: Data used for grooming benchmark. Number of videos (first column), and number of annotated frames (second and third columns). Table 2: Classifiers trained by JABS with their respective window sizes and F1 scores.

Supplementary file 2. Behavioral phenotypes definitions. Table 3: Summary of framewise behavioral phenotypes and their definitions. Each value corresponds to the total duration (in s) of the indicated behavior during the specified time window, averaged across all analyzed videos. Table 4: Behavioral phenotypes annotated by different annotators (A1, A2). Each phenotype measures a specific metric related to bouts of the indicated behavior during the first 55 min of the video averaged across all

the analyzed videos.

Supplementary file 3. Table of JABS features used for training behavioral classifiers. Table 5: List of JABS per-frame features.

Supplementary file 4. List of Mouse strains used in this study and their JAX stock numbers. Table 6: Mouse strains used in this study and their JAX stock numbers.

## Data availability

All behavior data is available through Harvard Dataverse. https://doi.org/10.7910/DVN/SAPNJG, https://doi.org/10.7910/DVN/RQYI04. All code is available through the Kumar Lab Github repositories (https://github.com/KumarLabJax/JABS-behavior-classifier copy archived at *Beane, 2026*; https://github.com/KumarLabJax/JABS-data-pipeline copy archived at *Geuther et al., 2026a*).

The following datasets were generated:

| Author(s) | Year | Dataset title | Dataset URL | Database and Identifier |
|---|---|---|---|---|
| Kumar V, Geuther B, Deats S | 2024 | The Jackson Laboratory (JAX) - Kumar Lab Mouse Strain Survey of Behavior in the Open Field Arena (Video Dataset) | https://doi.org/10.7910/DVN/SAPNJG | Harvard Dataverse, 10.7910/DVN/SAPNJG |
| Kumar V, Deats S | 2024 | The Jackson Laboratory (JAX) - Kumar Lab Mouse BxD Survey of Behavior in the Open Field Arena (Video Dataset) | https://doi.org/10.7910/DVN/RQYI04 | Harvard Dataverse, 10.7910/DVN/RQYI04 |

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
