## [Editor Report · eLife Assessment]

This **important** study presents JABS, an open-source platform that integrates hardware and user-friendly software for standardized mouse behavioral phenotyping. The work has practical implications for improving reproducibility and accessibility in behavioral neuroscience, especially for linking behavior to genetics across diverse mouse strains. The strength of evidence is **convincing**, with comprehensive validation of the platform's components and enthusiastic reviewer support.

---

## [Referee Report · Reviewer #1 (Public review)]

Summary:

This manuscript provides an open-source tool including hardware and software, and dataset to facilitate and standardize behavioral classification in laboratory mice. The hardware for behavioral phenotyping was extensively tested for safety. The software is GUI based facilitating the usage of this tool across the community of investigators that do not have a programming background. The behavioral classification tool is highly accurate, and the authors deposited a large dataset of annotations and pose tracking for many strains of mice. This tool has great potential for behavioral scientists that use mice across many fields, however there are many missing details that currently limit the impact of this tool and publication.

Strengths:

Software-hardware integration for facilitating cross-lab adaptation of the tool and minimizing the need to annotate new data for behavioral classification.

Data from many strains of mice was included in the classification and genetic analyses in this manuscript.

Large dataset annotated was deposited for the use of the community

GUI based software tool decreases barriers of usage across users with limited coding experience.

Weaknesses:

The GUI requires pose tracking for classification but, the software provided in JABS does not do pose tracking, so users must do pose tracking using a separate tool. The pose tracking quality directly impacts the classification quality, given that it is used for the feature calculation

Comments on revisions:

The authors addressed all my concerns.

---

## [Referee Report · Reviewer #2 (Public review)]

Summary:

This manuscript presents the JAX Animal Behavior System (JABS), an integrated mouse phenotyping platform that includes modules for data acquisition, behavior annotation, and behavior classifier training and sharing. The manuscript provides details and validation for each module, demonstrating JABS as a useful open-source behavior analysis tool that removes barriers to adopting these analysis techniques by the community. In particular, with the JABS-AI module users can download and deploy previously trained classifiers on their own data, or annotate their own data and train their own classifiers. The JABS-AI module also allows users to deploy their classifiers on the JAX strain survey dataset and receive an automated behavior and genetic report.

Strengths:

(1) The JABS platform addresses the critical issue of reproducibility in mouse behavior studies by providing an end-to-end system from rig setup to downstream behavioral and genetic analyses. Each step has clear guidelines, and the GUIs are an excellent way to encourage best practices for data storage, annotation, and model training. Such a platform is especially helpful for labs without prior experience in this type of analysis.

(2) A notable strength of the JABS platform is its reuse of large amounts of previously collected data at JAX Labs, condensing this into pretrained pose estimation models and behavioral classifiers. JABS-AI also provides access to the strain survey dataset through automated classifier analyses, allowing large-scale genetic screening based on simple behavioral classifiers. This has the potential to accelerate research for many labs by identifying particular strains of interest.

(3) The ethograph analysis will be a useful way to compare annotators/classifiers beyond the JABS platform.

Weaknesses:

(1) The manuscript contains many assertions that lack references in both the Introduction and Discussion. For example, in the Discussion, the assertion "published research demonstrates that keypoint detection models maintain robust performance despite the presence of headstages and recording equipment" lacks reference.

(2) The provided GUIs lower the barrier to entry for labs that are just starting to collect and analyze mouse open field behavior data. However, users must run pose estimation themselves outside of the provided GUIs, which introduces a key bottleneck in the processing pipeline, especially for users without strong programming skills. The authors have provided pretrained pose estimation models and an example pipeline, which is certainly to be commended, but I believe the impact of these tools could be greatly magnified by an additional pose estimation GUI (just for running inference, not for labeling/training).

(3) While the manuscript does a good job of laying out best practices, there is an opportunity to further improve reproducibility for users of the platform. The software seems likely to perform well with perfect setups that adhere to the JABS criteria, but it is very likely there will be users with suboptimal setups - poorly constructed rigs, insufficient camera quality, etc. It is important, in these cases, to give users feedback at each stage of the pipeline so they can understand if they have succeeded or not. Quality control (QC) metrics should be computed for raw video data (is the video too dark/bright? are there the expected number of frames? etc.), pose estimation outputs (do the tracked points maintain a reasonable skeleton structure; do they actually move around the arena?), and classifier outputs (what is the incidence rate of 1-3 frame behaviors? a high value could indicate issues). In cases where QC metrics are difficult to define (they are basically always difficult to define), diagnostic figures showing snippets of raw data or simple summary statistics (heatmaps of mouse location in the open field) could be utilized to allow users to catch glaring errors before proceeding to the next stage of the pipeline, or to remove data from their analyses if they observe critical issues.

Comments on revisions:

I thank the authors for taking the time to address my comments. They have provided a lot of important context in their responses. My only remaining recommendation is to incorporate more of this text into the manuscript itself, as this context will also be interesting/important for readers (and potential users) to consider. Specifically:

the quality control/user feedback features that have already been implemented (these are extremely important, and unfortunately, not standard practice in many labs)

top-down vs bottom-up imaging trade-offs (you make very good points!)

video compression, spatial and temporal resolution trade-offs

more detail on why the authors chose pose-based rather than pixel-based classifiers

I believe the proposed system can be extremely useful for behavioral neuroscientists, especially since the top-down freely moving mouse paradigm is one of the most ubiquitous in the field. Many labs have reinvented the wheel here, and as a field it makes sense to coalesce around a set of pipelines and best practices to accelerate the science we all want to do. I make the above recommendation with this in mind: bringing together (properly referenced) observations and experiences of the authors themselves, as well as others in the field, provides a valuable resource for the community. Obviously, the main thrust of the manuscript should be about the tools themselves; it should not turn into a review paper, so I'm just suggesting some additional sentences/references sprinkled throughout as motivation for why the authors made the choices that they did.

Intro typo: "one link in the chainDIY rigs"

---

## [Author Response]

The following is the authors’ response to the original reviews.

**Reviewer #1 (Public review):**
(1) The authors only report the quality of the classification considering the number of videos used for training, but not considering the number of mice represented or the mouse strain. Therefore, it is unclear if the classification model works equally well in data from all the mouse strains tested, and how many mice are represented in the classifier dataset and validation.

We agree that strain-level performance is critical for assessing generalizability. In the revision we now report per-strain accuracy and F1 for the grooming classifier, which was trained on videos spanning 60 genetically diverse strains (n = 1100 videos) and evaluated on the test set videos spanning 51 genetically diverse strains (n=153 videos). Performance is uniform across most strains (median F1 = 0.94, IQR = 0.899–0.956), with only modest declines in albino lines that lack contrast under infrared illumination; this limitation and potential remedies are discussed in the text. The new per-strain metrics are presented in the Supplementary figure (corresponding to Figure 4).

(2) The GUI requires pose tracking for classification, but the software provided in JABS does not do pose tracking, so users must do pose tracking using a separate tool. Currently, there is no guidance on the pose tracking recommendations and requirements for usage in JABS. The pose tracking quality directly impacts the classification quality, given that it is used for the feature calculation; therefore, this aspect of the data processing should be more carefully considered and described.

We have added a section to the methods describing how to use the pose estimation models used in JABS. The reviewer is correct that pose tracking quality will impact classification quality. We recommend that classifiers should only be re-used on pose files generated by the same pose models used in the behavior classifier training dataset. We hope that the combination of sharing classifier training data and making a more unified framework for developing and comparing classifiers will get us closer to having foundational behavior classification models that work in many environments. We also would like to emphasize that deviating from using our pose model will also likely hinder re-using our shared large datasets in JABS-AI (JABS1200, JABS600, JABS-BxD).

(3) Many statistical and methodological details are not described in the manuscript, limiting the interpretability of the data presented in Figures 4,7-8. There is no clear methods section describing many of the methods used and equations for the metrics used. As an example, there are no details of the CNN used to benchmark the JABS classifier in Figure 4, and no details of the methods used for the metrics reported in Figure 8.

We thank the reviewer for bringing this to our attention. We have added a methods section to the manuscript to address this concern. Specifically, we now provide: (1) improved citation visibility of the source of CNN experiments such that the reader can locate the architecture information, (2) mathematical formulations for all performance metrics (precision, recall, F1, …) with explicit equations; (3) detailed statistical procedures including permutation testing methods, power analysis and multiple testing corrections used throughout Figures 7-8. These additions facilitate reproducibility and proper interpretation of all quantitative results presented in the manuscript.

**Reviewer #2 (Public review):**
(1) The manuscript as written lacks much-needed context in multiple areas: what are the commercially available solutions, and how do they compare to JABS (at least in terms of features offered, not necessarily performance)? What are other open-source options?

JABS adds to a list of commercial and open source animal tracking platforms. There are several reviews and resources that cover these technologies. JABS covers hardware, behavior prediction, a shared resource for classifiers, and genetic association studies. We’re not aware of another system that encompasses all these components. Commercial packages such as EthoVision XT and HomeCage Scan give users a ready-made camera-plus-software solution that automatically tracks each mouse and reports simple measures such as distance travelled or time spent in preset zones, but they do not provide open hardware designs, editable behavior classifiers, or any genetics workflow. At the open-source end, the >100 projects catalogued on OpenBehavior and summarised in recent reviews (Luxem et al., 2023; Işık & Ünal 2023) usually cover only one link in the chain—DIY rigs, pose-tracking libraries (e.g., DeepLabCut, SLEAP) or supervised and unsupervised behaviour-classifier pipelines (e.g., SimBA, MARS, JAABA, B-SOiD, DeepEthogram). JABS provides an open source ecosystem that integrates all four: (i) top-down arena hardware with parts list and assembly guide; (ii) an active-learning GUI that produces shareable classifiers; (iii) a public web service that enables sharing of the trained classifier and applies any uploaded classifier to a large and diverse strain survey; and (iv) built-in heritability, genetic-correlation and GWAS reporting. We have added a concise paragraph in the Discussion that cites these resources and makes this end-to-end distinction explicit.

(2) How does the supervised behavioral classification approach relate to the burgeoning field of unsupervised behavioral clustering (e.g., Keypoint-MoSeq, VAME, B-SOiD)?

The reviewer raises an important point about the rapidly evolving landscape of automated behavioral analysis, where both supervised and unsupervised approaches offer complementary strengths for different experimental contexts. Unsupervised methods like Keypoint-MoSeq , VAME , and B-SOiD , which prioritize motif discovery from unlabeled data but may yield less precise alignments with expert annotations, as evidenced by lower F1 scores in comparative evaluations. Supervised approaches (like ours), by contrast, employ fully supervised classifiers to deliver frame-accurate, behavior-specific scores that align directly with experimental hypotheses. Ultimately, a pragmatic hybrid strategy, starting with unsupervised pilots to identify motifs and transitioning to supervised fine-tuning with minimal labels, can minimize annotation burdens and enhance both discovery and precision in ethological studies. This has been added in the discussion section of the manuscript.

(3) What kind of studies will this combination of open field + pose estimation + supervised classifier be suitable for? What kind of studies is it unsuited for? These are all relevant questions that potential users of this platform will be interested in.

This approach is suitable for a wide array of neuroscience, genetics, pharmacology, preclinical, and ethology studies. We have published in the domains of action detection for complex behaviors such as grooming, gait and posture, frailty, nociception, and sleep. We feel these tools are indispensable for modern behavior analysis.

(4) Throughout the manuscript, I often find it unclear what is supported by the software/GUI and what is not. For example, does the GUI support uploading videos and running pose estimation, or does this need to be done separately? How many of the analyses in Figures 4-6 are accessible within the GUI?

We have now clarified these. The JABS framework comprises two distinct GUI applications with complementary functionalities. The JABS-AL (active learning) desktop application handles video upload, behavioral annotation, classifier training, and inference -- it does not perform pose estimation, which must be completed separately using our pose tracking pipeline (https://github.com/KumarLabJax/mouse-tracking-runtime). If a user does not want to use our pose tracking pipeline, we have provided conversions through SLEAP to convert to our JABS pose format. The web-based GUI enables classifier sharing and cloud-based inference on our curated datasets (JABS600, JABS1200) and downstream behavioral statistics and genetic analyses (Figures 4-6). The JABS-AL application also supports CLI (command line interface) operation for batch processing. We have clarified these distinctions and provided a comprehensive workflow diagram in the revised Methods section.

(5) While the manuscript does a good job of laying out best practices, there is an opportunity to further improve reproducibility for users of the platform. The software seems likely to perform well with perfect setups that adhere to the JABS criteria, but it is very likely that there will be users with suboptimal setups - poorly constructed rigs, insufficient camera quality, etc. It is important, in these cases, to give users feedback at each stage of the pipeline so they can understand if they have succeeded or not. Quality control (QC) metrics should be computed for raw video data (is the video too dark/bright? are there the expected number of frames? etc.), pose estimation outputs (do the tracked points maintain a reasonable skeleton structure; do they actually move around the arena?), and classifier outputs (what is the incidence rate of 1-3 frame behaviors? a high value could indicate issues). In cases where QC metrics are difficult to define (they are basically always difficult to define), diagnostic figures showing snippets of raw data or simple summary statistics (heatmaps of mouse location in the open field) could be utilized to allow users to catch glaring errors before proceeding to the next stage of the pipeline, or to remove data from their analyses if they observe critical issues.

These are excellent suggestions that align with our vision for improving user experience and data quality assessment. We recognize the critical importance of providing users with comprehensive feedback at each stage of the pipeline to ensure optimal performance across diverse experimental setups. Currently, we provide end-users with tools and recommendations to inspect their own data quality. In our released datasets (Strain Survey OFA and BXD OFA), we provide video-level quality summaries for coverage of our pose estimation models.

For behavior classification quality control, we employ two primary strategies to ensure proper operation: (a) outlier manual validation and (b) leveraging known characteristics about behaviors. For each behavior that we predict on datasets, we manually inspect the highest and lowest expressions of this behavior to ensure that the new dataset we applied it to maintains sufficient similarity. For specific behavior classifiers, we utilize known behavioral characteristics to identify potentially compromised predictions. As the reviewer suggested, high incidence rates of 1-3 frame bouts for behaviors that typically last multiple seconds would indicate performance issues.

We currently maintain in-house post-processing scripts that handle quality control according to our specific use cases. Future releases of JABS will incorporate generalized versions of these scripts, integrating comprehensive QC capabilities directly into the platform. This will provide users with automated feedback on video quality, pose estimation accuracy, and classifier performance, along with diagnostic visualizations such as movement heatmaps and behavioral summary statistics.

**Reviewer #1 (Recommendations for the authors):**
(1) A weakness of this tool is that it requires pose tracking, but the manuscript does not detail how pose tracking should be done and whether users should expect that the data deposited will help their pose tracking models. There is no specification on how to generate pose tracking that will be compatible with JABS. The classification quality is directly linked to the quality of the pose tracking. The authors should provide more details of the requirements of the pose tracking (skeleton used) and what pose tracking tools are compatible with JABS. In the user website link, I found no such information. Ideally, JABS would be integrated with the pose tracking tool into a single pipeline. If that is not possible, then the utility of this tool relies on more clarity on which pose tracking tools are compatible with JABS.

The JABS ecosystem was deliberately designed with modularity in mind, separating the pose estimation pipeline from the active learning and classification app (JABS-AL) to offer greater flexibility and scalability for users working across diverse experimental setups. Our pose estimation pipeline is documented in detail within the new Methods subsection, outlining the steps to obtain JABS-compatible keypoints with our recommended runtime (https://github.com/KumarLabJax/mouse-tracking-runtime) and frozen inference models (https://github.com/KumarLabJax/deep-hrnet-mouse). This pipeline is an independent component within the broader JABS workflow, generating skeletonized keypoint data that are then fed into the JABS-AL application for behavior annotation and classifier training.

By maintaining this separation, users have the option to use their preferred pose tracking tools— such as SLEAP —while ensuring compatibility through provided conversion utilities to the JABS skeleton format. These details, including usage instructions and compatibility guidance, are now thoroughly explained in the newly added pose estimation subsection of our Methods section. This modular design approach ensures that users benefit from best-in-class tracking while retaining the full power and reproducibility of our active learning pipeline.

(2) The authors should justify why JAABA was chosen to benchmark their classifier. This tool was published in 2013, and there have been other classification tools (e.g., SIMBA) published since then.

We appreciate the reviewer’s suggestion regarding SIMBA. However, our comparisons to JAABA and a CNN are based on results from prior work (Geuther, Brian Q., et al. "Action detection using a neural network elucidates the genetics of mouse grooming behavior." Elife 10 (2021): e63207.), where both were used to benchmark performance on our publicly released dataset. In this study, we introduce JABS as a new approach and compare it against those established baselines. While SIMBA may indeed offer competitive performance, we believe the responsibility to demonstrate this lies with SIMBA’s authors, especially given the availability of our dataset for benchmarking.

(3) I had a lot of trouble understanding the elements of the data calculated in JABS vs outside of JABS. This should be clarified in the manuscript.(a) For example, it was not intuitive that pose tracking was required and had to be done separately from the JABS pipeline. The diagrams and figures should more clearly indicate that.(b) In section 2.5, are any of those metrics calculated by JABS? Another software GEMMA, but no citation is provided for this tool. This created ambiguity regarding whether this is an analysis that is separate from JABS or integrated into the pipeline.

We acknowledge the confusion regarding the delineation between JABS components and external tools, and we have comprehensively addressed this throughout the manuscript. The JABS ecosystem consists of three integrated modules: JABS-DA (data acquisition), JABS-AL (active learning for behavior annotation and classifier training), and JABS-AI (analysis and integration via web application). Pose estimation, while developed by our laboratory, operates as a preprocessing pipeline that generates the keypoint coordinates required for subsequent JABS classifier training and annotation workflows. We have now added a dedicated Methods subsection that explicitly maps each analytical step to its corresponding software component, clearly distinguishing between core JABS modules and external tools (such as GEMMA for genetic analysis). Additionally, we have provided proper citations and code repositories for all external pipelines to ensure complete transparency regarding the computational workflow and enable full reproducibility of our analyses.

(4) There needs to be clearer explanations of all metrics, methods, and transformations of the data reported.(a) There is very little information about the architecture of the classification model that JABS uses.(b) There are no details on the CNN used for comparing and benchmarking the classifier in JABS.(c) Unclear how the z-scoring of the behavioral data in Figure 7 was implemented.(d) There is currently no information on how the metrics in Figure 8 are calculated.

We have added a comprehensive Methods section that not only addresses the specific concerns raised above but provides complete methodological transparency throughout our study. This expanded section includes detailed descriptions of all computational architectures (including the JABS classifier and grooming benchmark models and metrics), statistical procedures and data transformations (including the z-scoring methodology for Figure 7), downstream genetic analysis (including all measures presented in Figure 8), and preprocessing pipelines.

(5) The authors talk about their datasets having visual diversity, but without seeing examples, it is hard to know what they mean by this visual diversity. Ideally, the manuscript would have a supplementary figure with a representation of the variety of setups and visual diversity represented in the datasets used to train the model. This is important so that readers can quickly assess from reading the manuscript if the pre-trained classifier models could be used with the experimental data they have collected.

The visual diversity of our training datasets has been comprehensively documented in our previous tracking work (https://www.nature.com/articles/s42003-019-0362-1), which systematically demonstrates tracking performance across mice with diverse coat colors (black, agouti, albino, gray, brown, nude, piebald), body sizes including obese mice, and challenging recording conditions with dynamic lighting and complex environments. Notably, Figure 3B in that publication specifically illustrates the robustness across coat colors and body shapes that characterize the visual diversity in our current classifier training data. To address the reviewer's concern and enable readers to quickly assess the applicability of our pre-trained models to their experimental data, we have now added this reference to the manuscript to ground our claims of visual diversity in published evidence.

(6) All figures have a lot of acronyms used that are not defined in the figure legend. This makes the figures really hard to follow. The figure legends for Figures 1,2, 7, and 9 did not have sufficient information for me to comprehend the figure shown.

We have fixed this in the manuscript.

(7) In the introduction, the authors talk about compression artifacts that can be introduced in camera software defaults. This is very vague without specific examples.

This is a complex topic that balances the size and quality of video data and is beyond the scope of this paper. We have carefully optimized this parameter and given the user a balanced solution. A more detailed blog post on compression artifacts can be found at our lab’s webpage (https://www.kumarlab.org/2018/11/06/brians-video-compression-tests/). We have also added a comment about keyframes shifting temporal features in the main manuscript.

(8) More visuals of the inside of the apparatus should be included as supplementary figures. For example, to see the IR LEDs surrounding the camera.

We have shared data from JABS as part of several papers including the tracking paper (Geuther et al 2019), grooming, gait and posture, mouse mass. We have also released entire datasets that as part of this paper (JABS1800, JABS-BXD). We also have step by step assembly guide that shows the location of the lights/cameras and other parts see Methods, JABS workflow guide, and this PowerPoint file in the GitHub repository (https://github.com/KumarLabJax/JABS-datapipeline/blob/main/Multi-day%20setup%20PowerPoint%20V3.pptx).

(9) Figure 2 suggests that you could have multiple data acquisition systems simultaneously. Do each require a separate computer? And then these are not synchronized data across all boxes?

Each JABS-DA unit has its own edge device (Nvidia Jetson). Each system (which we define as multiple JABS-DA areas associated with one lab/group) can have multiple recording devices (arenas). The system requires only 1 control portal (RPi computer) and can handle as many recording devices as needed (Nvidia computer w/ camera associated with each JABS-DA arena). To collect data, 1 additional computer is needed to visit the web control portal and initiate a recording session. Since this is a web portal, users can use any computer or a tablet. The recording devices are not strictly synchronized but can be controlled in a unified manner.

(10) The list of parts on GitHub seems incomplete; many part names are not there.

We thank referee for bringing this to our attention. We have updated the GitHub repository (and its README) which now links out to the design files.

(11) The authors should consider adding guidance on how tethers and headstages are expected to impact the use of JABS, as many labs would be doing behavioral experiments combined with brain measurements.

While our pose estimation model was not specifically trained on tethered animals, published research demonstrates that keypoint detection models maintain robust performance despite the presence of headstages and recording equipment. Once accurate pose coordinates are extracted, the downstream behavior classification pipeline operates independently of the pose estimation method and would remain fully functional. We recommend users validate pose estimation accuracy in their specific experimental setup, as the behavior classification component itself is agnostic to the source of pose coordinates.

**Reviewer #2 (Recommendations for the authors):**
(1) "Using software-defaults will introduce compression artifacts into the video and will affect algorithm performance." Can this be quantified? I imagine most of the performance hit comes from a decrease in pose estimation quality. How does a decrease in pose estimation quality translate to action segmentation? Providing guidelines to potential users (e.g., showing plots of video compression vs classifier performance) would provide valuable information for anyone looking to use this system (and could save many labs countless hours replicating this experiment themselves). A relevant reference for the effect of compression on pose estimation is Mathis, Warren 2018 (bioRxiv): On the inference speed and video-compression robustness of DeepLabCut.

Since our behavior classification approach depends on features derived from keypoint, changes in keypoint accuracy will affect behavior segmentation accuracy. We agree that it is important to try and understand this further, particularly with the shared bioRxiv paper investigating the effect of compression on pose estimation accuracy. Measuring the effect of compression on keypoint and behavior classification is a complex task to evaluate concisely, given the number of potential variables to inspect. To list a few variables that should be investigated are: discrete cosine transform quality (Mathis, Warren experiment), Frame Size (Mathis, Warren experiment), Keyframe Interval (new, unique to video data), inter-frame settings (new, unique to video data), behavior of interest, Pose models with compression-augmentation used in training (https://arxiv.org/pdf/1506.08316?) and type of CNN used (under active development). The simplest recommendation that we can make at this time is that we know compression will affect behavior predictions and that users should be cautious about using our shared classifiers on compressed video data. To show that we are dedicated in sharing these results as we run those experiments, in a related work (CV4Animals conference accepted paper (https://www.cv4animals.com/) and can be downloaded here https://drive.google.com/file/d/1UNQIgCUOqXQh3vcJbM4QuQrq02HudBLD/view) we have already begun to inspect how changing some factors affect behavior segmentation performance. In this work, we investigate the robustness of behavior classification across multiple behaviors using different keypoint subsets. Our findings in this work is that classifiers are relatively stable across different keypoint subsets. We are actively working on follow-up effort to investigate the effect of keypoint noise, CNN model architecture, and other factors we've listed above on behavior segmentation tasks.

(2) The analysis of inter-annotator variability is very interesting. I'm curious how these differences compare to two other types of variability:(a) intra-annotator variability; I think this is actually hard to quantify with the presented annotation workflow. If a given annotator re-annotated a set of videos, but using different sparse subsets of the data, it is not possible to disentangle annotator variability versus the effect of training models on different subsets of data. This can only be rigorously quantified if all frames are labeled in each video.

We propose an alternative approach to behavior classifier development in the text associated with Figure 3C. We do not advocate for high inter-annotator agreement since individual behavior experts have differing labeling style (an intuitive understanding of the behavior). Rather, we allow multiple classifiers for the same behavior and allow the end user to prioritize classifiers based on heritability of the behavior from a classifier.

(b) In lieu of this, I'd be curious to see the variability in model outputs trained on data from a single annotator, but using different random seeds or train/val splits of the data. This analysis would provide useful null distributions for each annotator and allow for more rigorous statistical arguments about inter-annotator variability.

JABS allows the user to use multiple classifiers (random forest, XGBoost). We do not expect the user to carry out hyperparameter tuning or other forms of optimization. We find that the major increase in performance comes from optimizing the size of the window features and folds of cross validation. However, future versions of JABS-AL could enable a complete hyper-parameter scan across seeds and data splits to obtain a null distribution for each annotator.

(c) I appreciate the open-sourcing of the video/pose datasets. The authors might also consider publicly releasing their pose estimation and classifier training datasets (i.e., data plus annotations) for use by method developers.

We thank the referee for acknowledging our commitment to open data sharing practices. Building upon our previously released strain survey dataset, we have now also made our complete classifier training resources publicly available, including the experimental videos, extracted pose coordinates, and behavioral annotations. The repository link has been added to the manuscript to ensure full reproducibility and facilitate community adoption of our methods.

(3) More thorough discussion on the limitations of the top-down vs bottom-up camera viewpoint; are there particular scientific questions that are much better suited to bottomup videos (e.g., questions about paw tremors, etc.).

Top-down imaging, bottom-up, and multi-view imaging have a variety of pros and cons. Generally speaking, multi-view imaging will provide the most accurate pose models but requires increased resources on both hardware setup as well as processing of data. Top-down provides the advantage of flexibility for materials, since the floor doesn’t need to be transparent. Additionally lighting and potential reflection with the bottom-up perspective. Since the paws are not occluded from the bottom-up perspective, models should have improved paw keypoint precision allowing the model to observe more subtle behaviors. However, the appearance of the arena floor will change over time as the mice defecate and urinate. Care must be taken to clean the arena between recordings to ensure transparency is maintained. This doesn’t impact top-down imaging that much but will occlude or distort from the bottom-up perspective. Additionally, the inclusion of bedding for longer recordings, which is required by IACUC, will essentially render bottom-up imaging useless because the bedding will completely obscure the mouse. Overall, while bottomup may provide a precision benefit that will greatly enhance subtle motion, top-down imaging is overall more robust for obtaining consistent imaging across large experiments for longer periods of time.

(4) More thorough discussion on what kind of experiments would warrant higher spatial or temporal resolution (e.g., investigating slight tremors in a mouse model of neurodegenerative disease might require this greater resolution).

This is an important topic that deserves its own perspective guide. We try to capture some of this in the paper on specifications. However, we only scratch the surface. Overall, there are tradeoffs between frame rate, resolution, color/monochrome, and compression. Labs have collected data at hundreds of frames per second to capture the kinetics of reflexive behavior for pain (AbdoosSaboor lab) or whisking behavior. Labs have also collected data a low 2.5 frames per second for tracking activity or centroid tracking (see Kumar et al PNAS). The data collection specifications are largely dependent on the behaviors being captured. Our rule of thumb is the Nyquist Limit, which states that the data capture rate needs to be twice that of the frequency of the event. For example, certain syntaxes of grooming occur at 7Hz and we need 14FPS to capture this data. JABS collects data at 30FPS, which is a good compromise between data load and behavior rate. We use 800x800 pixel resolution which is a good compromise to capture animal body parts while limiting data size. Thank you for providing the feedback that the field needs guidance on this topic. We will work on creating such guidance documents for video data acquisition parameters to capture animal behavior data for the community as a separate publication.

(5) References(a) Should add the following ref when JAABA/MARS are referenced: Goodwin et al.2024, Nat Neuro (SimBA)(b) Could also add Bohnslav et al. 2021, eLife (DeepEthogram).(c) The SuperAnimal DLC paper (Ye et al. 2024, Nature Comms) is relevant to the introduction/discussion as well.

We thank the referee for the suggestions. We have added these references.

(6) Section 2.2:While I appreciate the thoroughness with which the authors investigated environmental differences in the JABS arena vs standard wean cage, this section is quite long and eventually distracted me from the overall flow of the exposition; might be worth considering putting some of the more technical details in the methods/appendix.

These are important data for adopters of JABS to gain IACUC approval in their home institution. These committees require evidence that any new animal housing environment has been shown to be safe for the animals. In the development of JABS, we spent a significant amount of time addressing the JAX veterinary and IACUC concerns. Therefore, we propose that these data deserve to be in the main text.

(7) Section 2.3.1:(a) Should again add the DeepEthogram reference here(b) Should reference some pose estimation papers: DeepLabCut, SLEAP, Lightning Pose.

We thank the referee for the suggestions. We have added these references.

(c) "Pose based approach offers the flexibility to use the identified poses for training classifiers for multiple behaviors" - I'm not sure I understand why this wouldn't be possible with the pixel-based approach. Is the concern about the speed of model training? If so, please make this clearer.

The advantage lies not just in training speed, but in the transferability and generalization of the learned representations. Pose-based approaches create structured, low-dimensional latent embeddings that capture behaviorally relevant features which can be readily repurposed across different behavioral classification tasks, whereas pixel-based methods require retraining the entire feature extraction pipeline for each new behavior. Recent work demonstrates that pose-based models achieve greater data efficiency when fine-tuned for new tasks compared to pixel-based transfer learning approaches [1], and latent behavioral representations can be partitioned into interpretable subspaces that generalize across different experimental contexts [2]. While pixel-based approaches can achieve higher accuracy on specific tasks, they suffer from the "curse of dimensionality" (requiring thousands of pixels vs. 12 pose coordinates per frame) and lack the semantic structure that makes pose-based features inherently reusable for downstream behavioral analysis.

(1) Ye, Shaokai, et al. "SuperAnimal pretrained pose estimation models for behavioral analysis." Nature communications 15.1 (2024): 5165.

(2) Whiteway, Matthew R., et al. "Partitioning variability in animal behavioral videos using semi-supervised variational autoencoders." PLoS computational biology 17.9 (2021): e1009439.

(d) The pose estimation portion of the pipeline needs more detail. Do users use a pretrained network, or do they need to label their own frames and train their own pose estimator? If the former, does that pre-trained network ship with the software? Is it easy to run inference on new videos from a GUI or scripts? How accurate is it in compliant setups built outside of JAX? How long does it take to process videos?

We have added the guidance on pose estimation in the manuscript (section “2.3.1 Behavior annotation and classifier training” and in the methods section titled “Pose tracking pipeline”)

(e) The final paragraph describing how to arrive at an optimal classifier is a bit confusing - is this the process that is facilitated by the app, or is this merely a recommendation for best practices? If this is the process the app requires, is it indeed true that multiple annotators are required? While obviously good practice, I imagine there will be many labs that just want a single person to annotate, at least in the beginning prototyping stages. Will the app allow training a model with just a single annotator?

We have clarified this in the text.

(8) Section 2.5:(a) This section contained a lot of technical details that I found confusing/opaque, and didn't add much to my overall understanding of the system; sec 2.6 did a good job of clarifying why 2.5 is important. It might be worth motivating 2.5 by including the content of 2.6 first, and moving some of the details of 2.5 to the method/appendix.

We moved some of the technical details in section 2.5 to the methods section titled “Genetic analysis”. Furthermore, we have added few statements to motivate the need of genetic analysis and how the webapp can facilitate this (which is introduced in the section 2.6)

(9) Minor corrections:(a) Bottom of first page, "always been behavior quantification task" missing "a".(b) "Type" column in Table S2 is undocumented and unused (i.e., all values are the same); consider removing.(c) Figure 4B, x-axis: add units.(d) Page 8/9: all panel references to Figure S1 are off by one

We have fixed them in the updated manuscript.